# Breaking the Bandwidth Barrier:
# Geometrical Adaptive Entropy Estimation

**Weihao Gao**,* **Sewoong Oh**,† and **Pramod Viswanath**∗
University of Illinois at Urbana-Champaign
Urbana, IL 61801
{wgao9,swoh,pramodv}@illinois.edu

## Abstract

Estimators of information theoretic measures such as entropy and mutual information are a basic workhorse for many downstream applications in modern data science. State of the art approaches have been either geometric (nearest neighbor (NN) based) or kernel based (with a globally chosen bandwidth). In this paper, we combine both these approaches to design new estimators of entropy and mutual information that outperform state of the art methods. Our estimator uses local bandwidth choices of $k$-NN distances with a finite $k$, independent of the sample size. Such a local and data dependent choice improves performance in practice, but the bandwidth is vanishing at a fast rate, leading to a non-vanishing bias. We show that the asymptotic bias of the proposed estimator is *universal*; it is independent of the underlying distribution. Hence, it can be precomputed and subtracted from the estimate. As a byproduct, we obtain a unified way of obtaining *both* kernel and NN estimators. The corresponding theoretical contribution relating the asymptotic geometry of nearest neighbors to order statistics is of independent mathematical interest.

## 1 Introduction

Unsupervised representation learning is one of the major themes of modern data science; a common theme among the various approaches is to extract maximally "informative" features via *information-theoretic metrics* (entropy, mutual information and their variations) – the primary reason for the popularity of information theoretic measures is that they are invariant to one-to-one transformations and that they obey natural axioms such as data processing. Such an approach is evident in many applications, as varied as computational biology [11], sociology [20] and information retrieval [17], with the citations representing a mere smattering of recent works. Within mainstream machine learning, a systematic effort at unsupervised clustering and hierarchical information extraction is conducted in recent works of [25, 23]. The basic workhorse in all these methods is the computation of mutual information (pairwise and multivariate) from i.i.d. samples. Indeed, *sample-efficient estimation* of mutual information emerges as the central scientific question of interest in a variety of applications, and is also of fundamental interest to statistics, machine learning and information theory communities.

While these estimation questions have been studied in the past three decades (and summarized in [28]), the renewed importance of estimating information theoretic measures in a *sample-efficient* manner is persuasively argued in a recent work [2], where the authors note that existing estimators perform poorly in several key scenarios of central interest (especially when the high dimensional random variables are strongly related to each other). The most common estimators (featured in scientific

---

∗Coordinated Science Lab and Department of Electrical and Computer Engineering
†Coordinated Science Lab and Department of Industrial and Enterprise Systems Engineering

software packages) are nonparametric and involve $k$ nearest neighbor (NN) distances between the samples. The widely used estimator of mutual information is the one by Kraskov and Stögbauer and Grassberger [10] and christened the KSG estimator (nomenclature based on the authors, cf. [2]) – while this estimator works well in practice (and performs much better than other approaches such as those based on kernel density estimation procedures), it still suffers in high dimensions. The basic issue is that the KSG estimator (and the underlying differential entropy estimator based on nearest neighbor distances by Kozachenko and Leonenko (KL) [9]) does not take advantage of the fact that the samples could lie in a smaller dimensional subspace (more generally, manifold) despite the high dimensionality of the data itself. Such lower dimensional structures effectively act as boundaries, causing the estimator to suffer from what is known as boundary biases.

Ameliorating this deficiency is the central theme of recent works [3, 2, 16], each of which aims to improve upon the classical KL (differential) entropy estimator of [9]. A local SVD is used to heuristically improve the density estimate at each sample point in [2], while a local Gaussian density (with empirical mean and covariance weighted by NN distances) is heuristically used for the same purpose in [16]. Both these approaches, while inspired and intuitive, come with no theoretical guarantees (even consistency) and from a practical perspective involve delicate choice of key hyper parameters. An effort towards a systematic study is initiated in [3] which connects the aforementioned heuristic efforts of [2, 16] to the *local log-likelihood* density estimation methods [6, 15] from theoretical statistics.

The local density estimation method is a strong generalization of the traditional kernel density estimation methods, but requires a delicate normalization which necessitates the solution of certain integral equations (cf. Equation (9) of [15]). Indeed, such an elaborate numerical effort is one of the key impediments for the entropy estimator of [3] to be practically valuable. A second key impediment is that theoretical guarantees (such as consistency) can only be provided when the bandwidth is chosen globally (leading to poor sample complexity in practice) and consistency requires the bandwidth $h$ to be chosen such that $nh^d \to \infty$ and $h \to 0$, where $n$ is the sample size and $d$ is the dimension of the random variable of interest. More generally, it appears that a systematic application of local log-likelihood methods to estimate *functionals* of the unknown density from i.i.d. samples is missing in the theoretical statistics literature (despite local log-likelihood methods for regression and density estimation being standard textbook fare [29, 14]). We resolve each of these deficiencies in this paper by undertaking a comprehensive study of estimating the (differential) entropy and mutual information from i.i.d. samples using sample dependent bandwidth choices (typically *fixed k-NN* distances). This effort allows us to connect disparate threads of ideas from seemingly different arenas: NN methods, local log-likelihood methods, asymptotic order statistics and sample-dependent heuristic, but inspired, methods for mutual information estimation suggested in the work of [10].

**Main Results**: We make the following contributions.

1. **Density** estimation: Parameterizing the log density by a polynomial of degree $p$, we derive *simple closed form* expressions for the local log-likelihood maximization problem for the cases of $p \le 2$ for arbitrary dimensions, with Gaussian kernel choices. This derivation, posed as an exercise in [14, Exercise 5.2], significantly improves the computational efficiency upon similar endeavors in the recent efforts of [3, 16, 26].

2. **Entropy** estimation: Using resubstitution of the local density estimate, we derive a simple closed form estimator of the entropy using a sample dependent bandwidth choice (of $k$-NN distance, where $k$ is a *fixed* small integer independent of the sample size): this estimator outperforms state of the art entropy estimators in a variety of settings. Since the bandwidth is data dependent and vanishes too fast (because $k$ is fixed), the estimator has a bias, which we derive a closed form expression for and show that it is *independent* of the underlying distribution and hence can be easily corrected: this is our main theoretical contribution, and involves new theorems on asymptotic statistics of nearest neighbors generalizing classical work in probability theory [19], which might be of independent mathematical interest.

3. **Generalized** view: We show that seemingly very different approaches to entropy estimation – recent works of [2, 3, 16] and the classical work of fixed $k$-NN estimator of Kozachenko and Leonenko [9] – can all be cast in the local log-likelihood framework as specific kernel and *sample dependent bandwidth* choices. This allows for a unified view, which we theoretically justify by showing that resubstitution entropy estimation for *any* kernel choice using fixed $k$-NN distances as bandwidth involves a bias term that is *independent of the underlying*

*distribution* (but depends on the specific choice of kernel and parametric density family). Thus our work is a strict mathematical generalization of the classical work of [9].

4. **Mutual Information** estimation: The inspired work of [10] constructs a mutual information estimator that subtly altered (in a sample dependent way) the three KL entropy estimation terms, leading to superior empirical performance. We show that the underlying idea behind this change can be incorporated in our framework as well, leading to a novel mutual information estimator that combines the two ideas and outperforms state of the art estimators in a variety of settings.

In the rest of this paper we describe these main results, the sections organized in roughly the same order as the enumerated list.

## 2 Local likelihood density estimation (LLDE)

Given $n$ i.i.d. samples $X_1, \ldots, X_n$, estimating the unknown density $f_X(\cdot)$ in $\mathbb{R}^d$ is a very basic statistical task. Local likelihood density estimators [15, 6] constitute state of the art and are specified by a weight function $K : \mathbb{R}^d \to \mathbb{R}$ (also called a kernel), a degree $p \in \mathbb{Z}^+$ of the polynomial approximation, and the bandwidth $h \in \mathbb{R}$, and maximizes the local log-likelihood:

$$\mathcal{L}_x(f) \;=\; \sum_{j=1}^{n} K\left(\frac{X_j - x}{h}\right) \log f(X_j) - n \int K\left(\frac{u - x}{h}\right) f(u)\, du \;, \tag{1}$$

where maximization is over an exponential polynomial family, locally approximating $f(u)$ near $x$:

$$\log_e f_{a,x}(u) = a_0 + \langle a_1, u - x \rangle + \langle u - x, a_2(u - x) \rangle + \cdots + a_p[u - x, u - x, \ldots, u - x]\;, \tag{2}$$

parameterized by $a = (a_0, \ldots, a_p) \in \mathbb{R}^{1 \times d \times d^2 \times \cdots \times d^p}$, where $\langle \cdot, \cdot \rangle$ denotes the inner-product and $a_p[u, \ldots, u]$ the $p$-th order tensor projection. The *local likelihood density estimate* (LLDE) is defined as $\widehat{f}_n(x) = f_{\widehat{a}(x),x}(x) = e^{\widehat{a}_0(x)}$, where $\widehat{a}(x) \in \arg\max_a \mathcal{L}_x(f_{a,x})$. The maximizer is represented by a series of nonlinear equations, and does not have a closed form in general. We present below a few choices of the degrees and the weight functions that admit closed form solutions. Concretely, for $p = 0$, it is known that LDDE reduces to the standard Kernel Density Estimator (KDE) [15]:

$$\widehat{f}_n(x) \;=\; \frac{1}{n} \sum_{i=1}^{n} K\left(\frac{x - X_i}{h}\right) \Big/ \int K\left(\frac{u - x}{h}\right) du \;. \tag{3}$$

If we choose the step function $K(u) = \mathbb{I}(\|u\| \leq 1)$ with a local and data-dependent choice of the bandwidth $h = \rho_{k,x}$ where $\rho_{k,x}$ is the $k$-NN distance from $x$, then the above estimator recovers the popular $k$-NN density estimate as a special case, namely, for $C_d = \pi^{d/2}/\Gamma(d/2 + 1)$,

$$\widehat{f}_n(x) \;=\; \frac{\frac{1}{n} \sum_{i=1}^{n} \mathbb{I}(\|X_i - x\| \leq \rho_{k,x})}{\mathrm{Vol}\{u \in \mathbb{R}^d : \|u - x\| \leq \rho_{k,x}\}} \;=\; \frac{k}{n\, C_d\, \rho_{k,x}^d} \;. \tag{4}$$

For higher degree local likelihood, we provide simple closed form solutions and provide a proof in Section D. Somewhat surprisingly, this result has eluded prior works [16, 26] and [3] which specifically attempted the evaluation for $p = 2$. Part of the subtlety in the result is to critically use the fact that the parametric family (eg., the polynomial family in (2)) need not be normalized themselves; the local log-likelihood maximization ensures that the resulting density estimate is correctly normalized so that it integrates to 1.

**Proposition 2.1.** *[14, Exercise 5.2] For a degree $p \in \{1, 2\}$, the maximizer of local likelihood* (1) *admits a closed form solution, when using the Gaussian kernel $K(u) = e^{-\frac{\|u\|^2}{2}}$. In case of $p = 1$,*

$$\widehat{f}_n(x) \;=\; \frac{S_0}{n(2\pi)^{d/2}h^d} \exp\left\{-\frac{1}{2}\frac{1}{S_0^2}\|S_1\|^2\right\}\;, \tag{5}$$

*where $S_0 \in \mathbb{R}$ and $S_1 \in \mathbb{R}^d$ are defined for given $x \in \mathbb{R}^d$ and $h \in \mathbb{R}$ as*

$$S_0 \equiv \sum_{j=1}^{n} e^{-\frac{\|X_j - x\|^2}{2h^2}}\;, \qquad S_1 \equiv \sum_{j=1}^{n} \frac{1}{h}(X_j - x)\, e^{-\frac{\|X_j - x\|^2}{2h^2}}\;. \tag{6}$$

*In case of $p = 2$, for $S_0$ and $S_1$ defined as above,*

$$\widehat{f}_n(x) \;=\; \frac{S_0}{n(2\pi)^{d/2}h^d|\Sigma|^{1/2}} \exp\left\{ -\frac{1}{2}\frac{1}{S_0^2}S_1^T\Sigma^{-1}S_1 \right\}, \qquad (7)$$

*where $|\Sigma|$ is the determinant and $S_2 \in \mathbb{R}^{d\times d}$ and $\Sigma \in \mathbb{R}^{d\times d}$ are defined as*

$$S_2 \equiv \sum_{j=1}^{n} \frac{1}{h^2}(X_j - x)(X_j - x)^T\, e^{-\frac{\|X_j - x\|^2}{2h^2}}, \qquad \Sigma \equiv \frac{S_0 S_2 - S_1 S_1^T}{S_0^2}, \qquad (8)$$

*where it follows from Cauchy-Schwarz that $\Sigma$ is positive semidefinite.*

One of the major drawbacks of the KDE and $k$-NN methods is the increased bias near the boundaries. LLDE provides a principled approach to automatically correct for the boundary bias, which takes effect only for $p \geq 2$ [6, 21]. This explains the performance improvement for $p = 2$ in the figure below (left panel), and the gap increases with the correlation as boundary effect becomes more prominent. We use the proposed estimators with $p \in \{0, 1, 2\}$ to estimate the mutual information between two jointly Gaussian random variables with correlation $r$, from $n = 500$ samples, using resubstitution methods explained in the next sections. Each point is averaged over 100 instances.

In the right panel, we generate i.i.d. samples from a 2-dimensional Gaussian with correlation 0.9, and found local approximation $\widehat{f}(u - x^*)$ around $x^*$ denoted by the blue $*$ in the center. Standard $k$-NN approach fits a uniform distribution over a circle enclosing $k = 20$ nearest neighbors (red circle). The green lines are the contours of the degree-2 polynomial approximation with bandwidth $h = \rho_{20,x}$. The figure illustrates that $k$-NN method suffers from boundary effect, where it underestimates the probability by over estimating the volume in (4). However, degree-2 LDDE is able to correctly capture the local structure of the pdf, correcting for boundary biases.

Despite the advantages of the LLDE, it requires the bandwidth to be data independent and vanishingly small (sublinearly in sample size) for consistency almost everywhere – both of these are impediments to practical use since there is no obvious systematic way of choosing these hyperparameters. On the other hand, if we restrict our focus to *functionals* of the density, then both these issues are resolved: this is the focus of the next section where we show that the bandwidth can be chosen to be based on *fixed* $k$-NN distances and the resulting universal bias easily corrected.

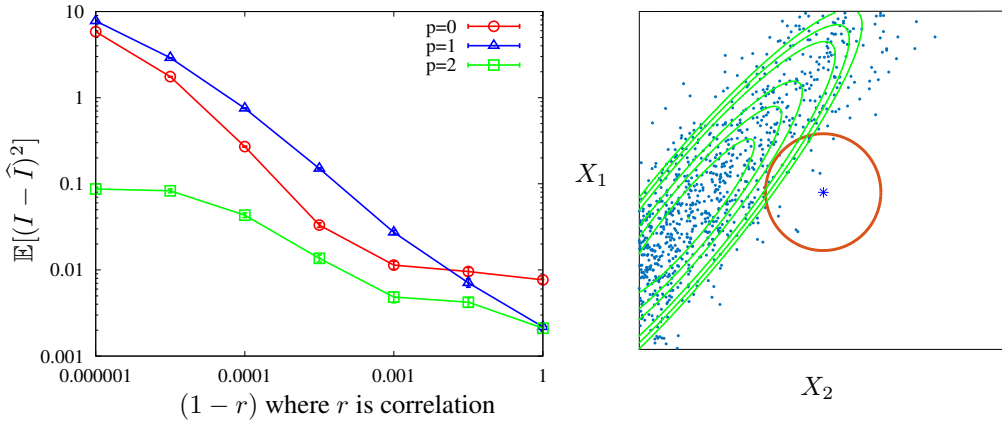

Figure 1: The boundary bias becomes less significant and the gap closes as correlation decreases for estimating the mutual information (left). Local approximation around the blue $*$ in the center. The degree-2 local likelihood approximation (contours in green) automatically captures the local structure whereas the standard $k$-NN approach (uniform distribution in red circle) fails (left).

## 3   $k$-LNN Entropy Estimator

We consider *resubstitution* entropy estimators of the form $\widehat{H}(x) = -(1/n)\sum_{i=1}^{n} \log \widehat{f}_n(X_i)$ and propose to use the local likelihood density estimator in (7) and a choice of bandwidth that is *local*

(varying for each point $x$) and *adaptive* (based on the data). Concretely, we choose, for each sample point $X_i$, the bandwidth $h_{X_i}$ to be the the distance to its $k$-th nearest neighbor $\rho_{k,i}$. Precisely, we propose the following $k$-Local Nearest Neighbor ($k$-LNN) entropy estimator of degree-2:

$$\widehat{H}_{k\mathrm{LNN}}^{(n)}(X) \;=\; -\frac{1}{n}\sum_{i=1}^{n}\left\{\log\frac{S_{0,i}}{n(2\pi)^{d/2}\rho_{k,i}^{d}|\Sigma_i|^{1/2}} - \frac{1}{2}\frac{1}{S_{0,i}^2}S_{1,i}^T\Sigma_i^{-1}S_{1,i}\right\} - B_{k,d}\,, \quad (9)$$

where subtracting $B_{k,d}$ defined in Theorem 1 removes the asymptotic bias, and $k \in \mathbb{Z}^+$ is the only hyper parameter determining the bandwidth. In practice $k$ is a small integer fixed to be in the range $4 \sim 8$. We only use the $\lceil \log n \rceil$ nearest subset of samples $\mathcal{T}_i = \{j \in [n]\ :\ j \neq i$ and $\|X_i - X_j\| \leq \rho_{\lceil \log n \rceil, i}\}$ in computing the quantities below:

$$S_{0,i} \equiv \sum_{j\in\mathcal{T}_{i,m}} e^{-\frac{\|X_j - X_i\|^2}{2\rho_{k,i}^2}}\,, \quad S_{1,i} \equiv \sum_{j\in\mathcal{T}_{i,m}} \frac{1}{\rho_{k,i}}(X_j - X_i)e^{-\frac{\|X_j - X_i\|^2}{2\rho_{k,i}^2}}\,,$$

$$S_{2,i} \equiv \sum_{j\in\mathcal{T}_{i,m}} \frac{1}{\rho_{k,i}^2}(X_j - X_i)(X_j - X_i)^T e^{-\frac{\|X_j - X_i\|^2}{2\rho_{k,i}^2}}\,, \quad \Sigma_i \equiv \frac{S_{0,i}S_{2,i} - S_{1,i}S_{1,i}^T}{S_{0,i}^2}\,. \quad (10)$$

The truncation is important for computational efficiency, but the analysis works as long as $m = O(n^{1/(2d)-\varepsilon})$ for any positive $\varepsilon$ that can be arbitrarily small. For a larger $m$, for example of $\Omega(n)$, those neighbors that are further away have a different asymptotic behavior. We show in Theorem 1 that the asymptotic bias is *independent* of the underlying distribution and hence can be *precomputed* and removed, under mild conditions on a twice continuously differentiable pdf $f(x)$ (cf. Lemma 3.1 below).

**Theorem 1.** *For $k \geq 3$ and $X_1, X_2, \ldots, X_n \in \mathbb{R}^d$ are i.i.d. samples from a twice continuously differentiable pdf $f(x)$, then*

$$\lim_{n\to\infty}\mathbb{E}[\widehat{H}_{k\mathrm{LNN}}^{(n)}(X)] \;=\; H(X)\,, \quad (11)$$

*where $B_{k,d}$ in (9) is a constant that only depends on $k$ and $d$. Further, if $\mathbb{E}[(\log f(X))^2] < \infty$ then the variance of the proposed estimator is bounded by $\mathrm{Var}[\widehat{H}_{k\mathrm{LNN}}^{(n)}(X)] = O((\log n)^2/n)$.*

This proves the $L_1$ and $L_2$ consistency of the $k$-LNN estimator; we relegate the proof to Section F for ease of reading the main part of the paper. The proof assumes Ansatz 1 (also stated in Section F), which states that a certain exchange of limit holds. As noted in [18], such an assumption is common in the literature on consistency of $k$-NN estimators, where it has been implicitly assumed in existing analyses of entropy estimators including [9, 5, 12, 27], without explicitly stating that such assumptions are being made. Our choice of a local adaptive bandwidth $h_{X_i} = \rho_{k,i}$ is crucial in ensuring that the asymptotic bias $B_{k,d}$ does not depend on the underlying distribution $f(x)$. This relies on a fundamental connection to the theory of asymptotic order statistics made precise in Lemma 3.1, which also gives the explicit formula for the bias below.

The main idea is that the empirical quantities used in the estimate (10) converge in large $n$ limit to similar quantities defined over order statistics. We make this intuition precise in the next section. We define order statistics over i.i.d. standard exponential random variables $E_1, E_2, \ldots, E_m$ and i.i.d. random variables $\xi_1, \xi_2, \ldots, \xi_m$ drawn uniformly (the Haar measure) over the unit sphere in $\mathbb{R}^d$, for a variable $m \in \mathbb{Z}^+$. We define for $\alpha \in \{0, 1, 2\}$,

$$\tilde{S}_\alpha^{(m)} \equiv \sum_{j=1}^{m} \xi_j^{(\alpha)} \frac{(\sum_{\ell=1}^{j} E_\ell)^\alpha}{(\sum_{\ell=1}^{k} E_\ell)^\alpha} \exp\left\{-\frac{(\sum_{\ell=1}^{j} E_\ell)^2}{2(\sum_{\ell=1}^{k} E_\ell)^2}\right\}\,, \quad (12)$$

where $\xi_j^{(0)} = 1$, $\xi_j^{(1)} = \xi_j \in \mathbb{R}^d$, and $\xi_j^{(2)} = \xi_j\xi_j^T \in \mathbb{R}^{d\times d}$, and let $\tilde{S}_\alpha = \lim_{m\to\infty}\tilde{S}_\alpha^{(m)}$ and $\widetilde{\Sigma} = (1/\tilde{S}_0)^2(\tilde{S}_0\tilde{S}_2 - \tilde{S}_1\tilde{S}_1^T)$. We show that the limiting $\tilde{S}_\alpha$'s are well-defined (in the proof of Theorem 1) and are directly related to the bias terms in the resubstitution estimator of entropy:

$$B_{k,d} = \mathbb{E}[\log(\sum_{\ell=1}^{k} E_\ell) + \frac{d}{2}\log 2\pi - \log C_d - \log \tilde{S}_0 + \frac{1}{2}\log|\widetilde{\Sigma}| + (\frac{1}{2\tilde{S}_0^2}\tilde{S}_1^T\widetilde{\Sigma}^{-1}\tilde{S}_1)]\,. \quad (13)$$

In practice, we propose using a fixed small $k$ such as five. For $k \leq 3$ the estimator has a very large variance, and numerical evaluation of the corresponding bias also converges slowly. For some typical choices of $k$, we provide approximate evaluations below, where $0.0183(\pm 6)$ indicates empirical mean $\mu = 183 \times 10^{-4}$ with confidence interval $6 \times 10^{-4}$. In these numerical evaluations, we truncated the summation at $m = 50,000$. Although we prove that $B_{k,d}$ converges in $m$, in practice, one can choose $m$ based on the number of samples and $B_{k,d}$ can be evaluated for that $m$.

**Theoretical contribution**: Our key technical innovation is a fundamental connection between nearest neighbor statistics and asymptotic order statistics, stated below as Lemma 3.1: we show that the (normalized) distances $\rho_{\ell,i}$'s jointly converge to the standardized uniform order statistics and the directions $(X_{j_\ell} - X_i)/\|X_{j_\ell} - X_i\|$'s converge to independent uniform distribution (Haar measure) over the unit sphere.

| | | $k$ | | | | | |
|---|---|---|---|---|---|---|---|
| | | 4 | 5 | 6 | 7 | 8 | 9 |
| $d$ | 1 | -0.0183($\pm$6) | -0.0233($\pm$6) | -0.0220($\pm$4) | -0.0200($\pm$4) | -0.0181($\pm$4) | -0.0171($\pm$3) |
| | 2 | -0.1023($\pm$5) | -0.0765($\pm$4) | -0.0628($\pm$4) | -0.0528($\pm$3) | -0.0448($\pm$3) | -0.0401($\pm$3) |

Table 1: Numerical evaluation of $B_{k,d}$, via sampling $1,000,000$ instances for each pair $(k,d)$.

Conditioned on $X_i = x$, the proposed estimator uses nearest neighbor statistics on $Z_{\ell,i} \equiv X_{j_\ell} - x$ where $X_{j_\ell}$ is the $\ell$-th nearest neighbor from $x$ such that $Z_{\ell,i} = ((X_{j_\ell} - X_i)/\|X_{j_\ell} - X_i\|)\rho_{\ell,i}$. Naturally, all the techniques we develop in this paper generalize to any estimators that depend on the nearest neighbor statistics $\{Z_{\ell,i}\}_{i,\ell \in [n]}$ – and the value of such a general result is demonstrated later (in Section 4) when we evaluate the bias in similarly inspired entropy estimators [2, 3, 16, 9].

**Lemma 3.1.** *Let $E_1, E_2, \ldots, E_m$ be i.i.d. standard exponential random variables and $\xi_1, \xi_2, \ldots, \xi_m$ be i.i.d. random variables drawn uniformly over the unit $(d-1)$-dimensional sphere in $d$ dimensions, independent of the $E_i$'s. Suppose $f$ is twice continuously differentiable and $x \in \mathbb{R}^d$ satisfies that there exists $\varepsilon > 0$ such that $f(a) > 0$, $\|\nabla f(a)\| = O(1)$ and $\|H_f(a)\| = O(1)$ for any $\|a - x\| < \varepsilon$. Then for any $m = O(\log n)$, we have the following convergence conditioned on $X_i = x$:*

$$\lim_{n \to \infty} d_{\mathrm{TV}}((c_d n f(x))^{1/d}(Z_{1,i}, \ldots, Z_{m,i}), (\xi_1 E_1^{1/d}, \ldots, \xi_m(\sum_{\ell=1}^{m} E_\ell)^{1/d})) = 0 . \quad (14)$$

*where $d_{\mathrm{TV}}(\cdot, \cdot)$ is the total variation and $c_d$ is the volume of unit Euclidean ball in $\mathbb{R}^d$.*

**Empirical contribution**: Numerical experiments suggest that the proposed estimator outperforms state-of-the-art entropy estimators, and the gap increases with correlation. The idea of using $k$-NN distance as bandwidth for entropy estimation was originally proposed by Kozachenko and Leonenko in [9], and is a special case of the $k$-LNN method we propose with degree 0 and a step kernel. We refer to Section 4 for a formal comparison. Another popular resubstitution entropy estimator is to use KDE in (3) [7], which is a special case of the $k$-LNN method with degree 0, and the Gaussian kernel is used in simulations. As comparison, we also study a new estimator [8] based on von Mises expansion (as opposed to simple re-substitution) which has an improved convergence rate in the large sample regime. We relegate simulation results to Section. B in the appendix.

## 4    Universality of the $k$-LNN approach

In this section, we show that Theorem 1 holds universally for a general family of entropy estimators, specified by the choice of $k \in \mathbb{Z}^+$, degree $p \in \mathbb{Z}^+$, and a kernel $K : \mathbb{R}^d \to \mathbb{R}$, thus allowing a unified view of several seemingly disparate entropy estimators [9, 2, 3, 16]. The template of the entropy estimator is the following: given $n$ i.i.d. samples, we first compute the local density estimate by maximizing the local likelihood (1) with bandwidth $\rho_{k,i}$, and then resubstitute it to estimate entropy: $\widehat{H}_{k,p,K}^{(n)}(X) = -(1/n)\sum_{i=1}^{n} \log \widehat{f}_n(X_i)$.

**Theorem 2.** *For the family of estimators described above, under the hypotheses of Theorem 1, if the solution to the maximization $\widehat{a}(x) = \arg\max_a \mathcal{L}_x(f_{a,x})$ exists for all $x \in \{X_1, \ldots, X_n\}$, then for any choice of $k \geq p + 1$, $p \in \mathbb{Z}^+$, and $K : \mathbb{R}^d \to \mathbb{R}$, the asymptotic bias is independent of the underlying distribution:*

$$\lim_{n \to \infty} \mathbb{E}[\widehat{H}_{k,p,K}^{(n)}(X)] = H(X) + \widetilde{B}_{k,p,K,d} , \quad (15)$$

*for some constant $\widetilde{B}_{k,d,p,K}$ that only depends on $k, p, K$ and $d$.*

We provide a proof in Section G. Although in general there is no simple analytical characterization of the asymptotic bias $\widetilde{B}_{k,p,K,d}$ it can be readily numerically computed: since $\widetilde{B}_{k,p,K,d}$ is independent of the underlying distribution, one can run the estimator over i.i.d. samples from *any* distribution and numerically approximate the bias for *any* choice of the parameters. However, when the maximization $\widehat{a}(x) = \arg\max_a \mathcal{L}_x(f_{a,x})$ admits a closed form solution, as is the case with proposed $k$-LNN, then $\widetilde{B}_{k,p,K,d}$ can be characterized explicitly in terms of uniform order statistics.

This family of estimators is general: for instance, the popular KL estimator is a special case with $p = 0$ and a step kernel $K(u) = \mathbb{I}(\|u\| \leq 1)$. [9] showed (in a remarkable result at the time) that the asymptotic bias is independent of the dimension $d$ and can be computed exactly to be $\log n - \psi(n) + \psi(k) - \log k$ and $\psi(k)$ is the digamma function defined as $\psi(x) = \Gamma^{-1}(x)d\Gamma(x)/dx$. The dimension independent nature of this asymptotic bias term (of $O(n^{-1/2})$ for $d = 1$ in [24, Theorem 1] and $O(n^{-1/d})$ for general $d$ in [4]) is special to the choice of $p = 0$ and the step kernel; we explain this in detail in Section G, later in the paper. Analogously, the estimator in [2] can be viewed as a special case with $p = 0$ and an ellipsoidal step kernel.

## 5  $k$-LNN Mutual information estimator

Given an entropy estimator $\widehat{H}_{\text{KL}}$, mutual information can be estimated: $\widehat{I}_{3\text{KL}} = \widehat{H}_{\text{KL}}(X) + \widehat{H}_{\text{KL}}(Y) - \widehat{H}_{\text{KL}}(X,Y)$. In [10], Kraskov and Stögbauer and Grassberger introduced $\widehat{I}_{\text{KSG}}(X;Y)$ by coupling the choices of the bandwidths. The joint entropy is estimated in the usual way, but for the marginal entropy, instead of using $k$NN distances from $\{X_j\}$, the bandwidth $h_{X_i} = \rho_{k,i}(X,Y)$ is chosen, which is the $k$ nearest neighbor distance from $(X_i, Y_i)$ for the joint data $\{(X_j, Y_j)\}$. Consider $\widehat{I}_{3\text{LNN}}(X;Y) = \widehat{H}_{k\text{LNN}}(X) + \widehat{H}_{k\text{LNN}}(Y) - \widehat{H}_{k\text{LNN}}(X,Y)$. Inspired by [10], we introduce the following novel mutual information estimator we denote by $\widehat{I}_{\text{LNN}-\text{KSG}}(X;Y)$. where for the joint $(X,Y)$ we use the LNN entropy estimator we proposed in (9), and for the marginal entropy we use the bandwidth $h_{X_i} = \rho_{k,i}(X,Y)$ coupled to the joint estimator. Empirically, we observe $\widehat{I}_{\text{KSG}}$ outperforms $\widehat{I}_{3\text{KL}}$ everywhere, validating the use of correlated bandwidths. However, the performance of $\widehat{I}_{\text{LNN}-\text{KSG}}$ is similar to $\widehat{I}_{3\text{LNN}}$–sometimes better and sometimes worse.

**Empirical Contribution**: Numerical experiments show that for most regimes of correlation, both 3LNN and LNN-KSG outperforms other state-of-the-art estimators, and the gap increases with correlation $r$. In the large sample limit, all estimators find the correct mutual information, but both LNN and LNN-KSG are significantly more robust compared to other approaches. Mutual information estimators have been recently proposed in [2, 3, 16] based on local likelihood maximization. However, they involve heuristic choices of hyper-parameters or solving elaborate optimization and numerical integrations, which are far from being easy to implement. Simulation results can be found in Section. C in the appendix.

## 6  Breaking the bandwidth barrier

While $k$-NN distance based bandwidth are routine in practical usage [21], the main finding of this work is that they also turn out to be the "correct" mathematical choice for the purpose of asymptotically unbiased estimation of an integral functional such as the entropy: $-\int f(x) \log f(x)$; we briefly discuss the ramifications below. Traditionally, when the goal is to estimate $f(x)$, it is well known that the bandwidth should satisfy $h \to 0$ and $nh^d \to \infty$, for KDEs to be consistent. As a rule of thumb, $h = 1.06\widehat{\sigma}n^{-1/5}$ is suggested when $d = 1$ where $\widehat{\sigma}$ is the sample standard deviation [29, Chapter 6.3]. On the other hand, when estimating entropy, as well as other integral functionals, it is known that resubstitution estimators of the form $-(1/n)\sum_{i=1}^n \log \widehat{f}(X_i)$ achieve variances scaling as $O(1/n)$ independent of the bandwidth [13]. This allows for a bandwidth as small as $O(n^{-1/d})$.

The bottleneck in choosing such a small bandwidth is the bias, scaling as $O(h^2 + (nh^d)^{-1} + E_n)$ [13], where the lower order dependence on $n$, dubbed $E_n$, is generally not known. The barrier in choosing a *global* bandwidth of $h = O(n^{-1/d})$ is the strictly positive bias whose value depends on the unknown distribution and cannot be subtracted off. However, perhaps surprisingly, the proposed local and

adaptive choice of the $k$-NN distance admits an asymptotic bias that is independent of the unknown underlying distribution. Manually subtracting off the non-vanishing bias gives an asymptotically unbiased estimator, with a potentially faster convergence as numerically compared below. Figure 2 illustrates how $k$-NN based bandwidth significantly improves upon, say a rule-of-thumb choice of $O(n^{-1/(d+4)})$ explained above and another choice of $O(n^{-1/(d+2)})$. In the left figure, we use the setting from Figure 3 (right) but with correlation $r = 0.999$. On the right, we generate $X \sim \mathcal{N}(0,1)$ and $U$ from uniform $[0, 0.01]$ and let $Y = X + U$ and estimate $I(X;Y)$. Following recent advances in [12, 22], the proposed local estimator has a potential to be extended to, for example, Renyi entropy, but with a multiplicative bias as opposed to additive.

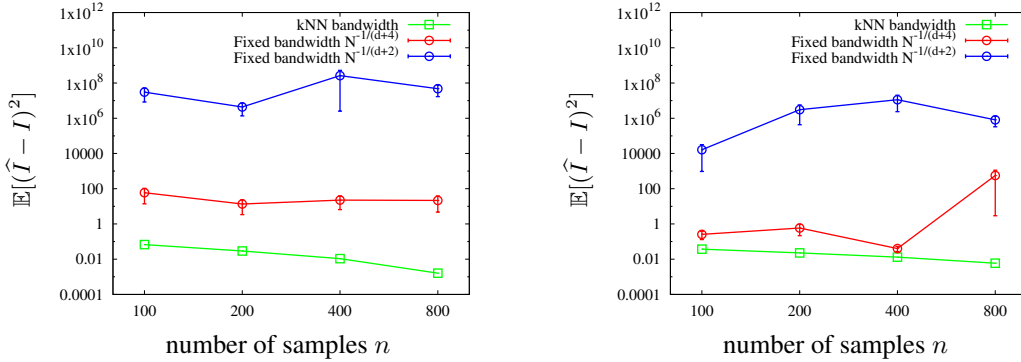

Figure 2: Local and adaptive bandwidth significantly improves over rule-of-thumb fixed bandwidth.

# Acknowledgement

This work is supported by NSF SaTC award CNS-1527754, NSF CISE award CCF-1553452, NSF CISE award CCF-1617745. We thank the anonymous reviewers for their constructive feedback.

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
