[Supplementary Material · local_supp.pdf]

# Supplementary material

## A  Discussion

The topic of estimation of an integral functional of an unknown density from i.i.d. samples is a classical one in statistics and we tie together a few pertinent topics from the literature in the context of the results of this manuscript.

### A.1  Uniform order statistics and NN distances

The expression for the asymptotic bias in (13) which is independent of the underlying distribution forms the main result of this paper and crucially depends on Lemma 3.1. Precisely, the lemma implies that the quantities $S_i$'s in (10) converge in distribution to $\tilde{S}_i$'s in (12). There are two parts to this convergence result: the nearest neighbor distances converge to uniform order statistics and the directions to those nearest neighbors converge independently to Haar measures on the unit sphere. The former has been extensively studied, for example see [29] for a survey of results. The latter is a new result that we state in Lemma 3.1, and proved in Section E. Intuitively, assuming smoothness, the probability density $f_X$ in the neighborhood of a sample $X_i$ (as defined by the distance to the $k$-th nearest neighbor) converges to a uniform distribution over a ball (of radius decreasing at the rate $\rho_{k,i} = \Theta(n^{-1/d})$), as more samples are collected. The nearest neighbor distances and directions converge to those from the uniform distribution over the ball, and Lemma 3.1 makes this intuition precise for the nearest $m$ neighbors up to $m = O(n^{1/(2d)-\epsilon})$ with any arbitrarily small but positive $\varepsilon$.

Only the convergence analysis of the distances, and not the directions, is required for traditional $k$-NN based estimators, such as the entropy estimator of [15]. In the seminal paper, [15] introduced *resubstitution* entropy estimators of the form $\widehat{H}(X) = -(1/n)\sum_{i=1}^{n} \log \widehat{f}_n(X_i)$ with $\widehat{f}_n(x) = k/(n\, C_d\, \rho_{k,x}^d)$ (as defined in (4)). This $k$-NN estimator has a non-vanishing asymptotic bias, which was computed as $B_{k,d} = (\psi(k) - \log(k))$ with the digamma function $\psi(\cdot)$ and was suggested to be manually removed. For $k = 1$ this was proved in the original paper of [15], which later was extended in [32, 9] to general $k$. This mysterious bias term $B_{k,d} = (\psi(k) - \log(k))$ whose original proofs in [15, 32, 9] provided little explanation for, can be alternatively proved with both rigor and intuition by making connections to uniform order statistics. For a special case of $k = 1$, with extra assumptions on the support being compact, such an elegant proof is provided in [2, Theorem 7.1] which explicitly applies the convergence of the nearest neighbor distance to uniform order statistics. Namely,

$$\mathbb{E}[\widehat{H}(X)] = \mathbb{E}\Big[ -\frac{1}{n}\sum_{i=1}^{n} \log\Big(\frac{k}{n\, C_d\, \rho_{k,X_i}^d}\Big)\Big]$$

$$\rightarrow \quad \mathbb{E}\Big[ -\log \frac{k\, f(X_i)}{\sum_{j=1}^{k} E_j}\Big] = H(X) + \psi(k) - \log(k)\,,$$

where the asymptotic expression follows from $C_d\, n\, f(x)\rho_{k,x}^d \rightarrow \sum_{j=1}^{k} E_j$ as shown, for example, in Lemma 3.1 and we used $\mathbb{E}[\log \sum_{j=1}^{k} E_j] = \psi(k)$, where $\psi(k) =$ is the digamma function defined as $\psi(x) = \Gamma^{-1}(x)d\Gamma(x)/dx$ and for large $x$ it is approximately $\log(x)$ up to $O(1/x)$, i.e. $\psi(x) = \log x - 1/(2x) + o(1/x)$. Note that this only requires the convergence of the distance and not the direction. Inspired by this modern approach, we extend such a connection in Lemma 3.1 to prove consistency of our estimator.

### A.2  Convergence rate of the bias

Establishing the convergence rate of the KL estimator is a challenging problem, and is not quite resolved despite work over the past three decades. The $O(1/n)$ convergence rate of the *variance* is established in [3, 18, 2, 4] under various assumptions. Establishing the convergence rate of the *bias* is more challenging. It has been first studied in [10, 11], where root-$n$ consistency is shown in 1-dimension with bounded support and assuming $f(x)$ is bounded below. [36] is the first to prove a root mean squared error convergence rate of $O(1/\sqrt{n})$ for general densities with unbounded support in 1-dimension and exponentially decaying tail, such as the Gaussian density. These assumptions are relaxed in [5], where zeroes and fat tails are allowed in $f(x)$. In general $d$-dimensions, [8, 33]

prove bounds on the convergence rate of the bias for finite $k = O(1)$, and [24, 1] for $k = \Omega(\log n)$. Establishing the convergence rate for the bias of the proposed local estimator is an interesting open problem – it is interesting to see if the superior empirical performance of the local estimator is captured in the asymptotics of rate of convergence of the bias.

It is intuitive that kernel density estimators can capture the structure in the distribution if the distribution lies on a lower dimensional manifold. This is made precise in [27], which also shows improved convergence rates for distributions whose support is on low dimensional manifolds. However, the estimator in [27] critically uses the geodesic distances between the sample points on the manifold. Given that the proposed estimators fit distributions locally, a concrete question of interest is whether such an improvement can be achieved *without* such an explicit knowledge of the geodesic distances, i.e., whether the local estimators automatically adapt to underlying lower dimensional structures.

### A.3 Ensemble estimators

Recent works [34, 25, 26, 1] have proposed ensemble estimators, which use known estimators based on kernel density estimators and $k$-NN methods and construct a new estimate by taking the weighted linear combination of those methods with varying bandwidth or $k$, respectively. With a proper choice of the weights, which can be computed analytically by solving a simple linear program, a boosting of the convergence rate can be achieved. The key property that allows the design of such ensemble estimators is that the leading terms (in terms of the sample size $n$) of the bias have a multiplicative constant that only depends on the unknown distribution. An intuitive explanation for this phenomenon is provided in [1] in the context of $k$-NN methods; it is interesting to explore if such a phenomenon continues in the $k$-LNN scenario studied in this paper. Such a study would potentially lead to ensemble-based estimators in the local setting and also naturally allow a careful understanding of the rate of convergence of the bias term.

## B Simulation Results in Section. 3

In Figure 3 (left), we draw 100 samples i.i.d. from two standard Gaussian random variables with correlation $r$, and plot resulting mean squared error averaged over 100 instances. The ground truth, in this case is $H(X) = \log(2\pi e) + 0.5\log(1 - r^2)$. On the right, we repeat the same simulation for fixed $r = 0.99999$ and varying number of samples and $m = 7\log_e n$.

Figure 3: Degree-2 $k$-LNN outperforms other state-of-the-art estimators for entropy estimation.

In Figure 4, we repeat the same simulation for 6 standard Gaussian random variables with $\mathrm{Cov}(X_1, X_2) = \mathrm{Cov}(X_3, X_4) = \mathrm{Cov}(X_5, X_6) = r$ and $\mathrm{Cov}(X_i, X_j) = 0$ for other pairs $(i, j)$. On the left, we draw 100 i.i.d. samples with various $r$. We plot resulting mean squared error averaged over 100 instances. The ground truth is $H(X) = 3\log(2\pi e) + 1.5\log(1 - r^2)$. On the right, we repeat the same simulation for fixed $r = 0.99999$ and varying number of samples and $m = 7\log_e n$.

In Figure 5 (left), we draw 100 samples i.i.d. from a mixture of two joint Gaussian distributions with zero mean and covariance $\begin{pmatrix} 1 & r \\ r & 1 \end{pmatrix}$ and $\begin{pmatrix} 1 & -r \\ -r & 1 \end{pmatrix}$, respectively, and plot resulting average estimate

Figure 4: Degree-2 $k$-LNN outperforms other state-of-the-art estimators for high-dimensional entropy estimation.

over 100 instances. Here we plot an upper bound of the ground truth $H(X) \leq \log(2) + \log(2\pi e) + 0.5 \log(1 - r^2)$ for $r \geq 0.9$. On the right, we repeat the same simulation for fixed $r = 0.99999$ and varying number of samples and $m = 7 \log_e n$.

Figure 5: Degree-2 $k$-LNN outperforms other state-of-the-art estimators for non-Gaussian entropy estimation.

## C  Simulation Results in Section. 5

In Figure 6 (left), we estimate mutual information under the same setting as in Figure 3 (left). For most regimes of correlation $r$, both 3LNN and LNN-KSG outperforms other state-of-the-art estimators. The gap increases with correlation $r$. On the right, we draw i.i.d. samples from two random variables $X$ and $Y$, where $X$ is uniform over $[0, 1]$ and $Y = X + U$, where $U$ is uniform over $[0, 0.01]$ independent of $X$. In the large sample limit, all estimators find the correct mutual information. The plot show how sensitive the estimates are, in the small sample regime. Both LNN and LNN-KSG are significantly more robust compared to other approaches.

In Figure 7, we test the mutual information estimators for $Y = f(X) + U$, where $X$ is uniformly distributed over $[0, 1]$ and $U$ is uniformly distributed over $[0, \theta]$, independent of $X$, for some noise level $\theta$. Similar simulation were studied in [7]. We draw 2500 i.i.d. sample points for each relationship. The plot show that for small noise level $\theta$, i.e., near-functional related random variables, our proposed estimators $\widehat{I}_{3LNN}$ and $\widehat{I}_{LNN-KSG}$ perform much better than 3KL and KSG estimators. Also our proposed estimators can handle both linear and nonlinear functional relationships.

In Figure 8, we test our estimators on linear and nonlinear relationships for both low-dimensional ($D = 2$) and high-dimensional ($D = 5$). Here $X_i$'s are uniformly distributed over $[0, 1]$ and $U$ is

Figure 6: Proposed $\widehat{I}_{\mathrm{LNN-KSG}}$ and $\widehat{I}_{\mathrm{3LNN}}$ outperform other state-of-the-art estimators.

Figure 7: Functional relationship test for mutual information estimators. Proposed $\widehat{I}_{\mathrm{LNN-KSG}}$ and $\widehat{I}_{\mathrm{3LNN}}$ outperform other state-of-the-art estimators.

uniformly distributed over $[-3^8/2, 3^8/2]$, independently of $X_i$'s. Similar simulation were studied in [6]. We can see that our estimators $\widehat{I}_{3LNN}$ and $\widehat{I}_{LNN-KSG}$ converges much faster than $\hat{I}_{3KL}$ and $\hat{I}_{KSG}$.

## D Proof of proposition 2.1

We first prove the derivation of the LLDE with degree $p = 2$ in Equation (7). The gradient of the local likelihood evaluated at the maximizer is zero [21], which gives a computational tool for finding

Figure 8: Estimated Mutual Information of low/high-dimensional relationships

the maximizer:

$$\frac{1}{n}\sum_{j=1}^{n} K(\frac{X_j - x}{h}) = \int K(\frac{u - x}{h})e^{a_0 + a_1^T(u-x) + (u-x)^T a_2(u-x)}du , \tag{16}$$

$$\frac{1}{n}\sum_{j=1}^{n} \frac{X_j - x}{h}K(\frac{X_j - x}{h}) = \int \frac{u - x}{h}K(\frac{u - x}{h})e^{a_0 + a_1^T(u-x) + (u-x)^T a_2(u-x)}du , \tag{17}$$

$$\frac{1}{n}\sum_{j=1}^{n} \frac{(X_j - x)(X_j - x)^T}{h^2}K(\frac{X_j - x}{h})$$
$$= \int \frac{(u - x)(u - x)^T}{h^2}K(\frac{u - x}{h})e^{a_0 + a_1^T(u-x) + (u-x)^T a_2(u-x)}du , \tag{18}$$

where $K(x) = \exp\{-\|x\|^2/2\}$ is the Gaussian kernel. Notice that the left-hand side of the equations are $S_0/n$, $S_1/n$ and $S_2/n$, respectively. The RHS can be written in closed forms as:

$$\frac{1}{n}S_0 = (2\pi)^{d/2}|M|^{-1/2}e^{a_0 + \frac{1}{2}a_1^T M^{-1} a_1} , \tag{19}$$

$$\frac{1}{n}S_1 = \frac{1}{nh}S_0 M^{-1} a_1 , \tag{20}$$

$$\frac{1}{n}S_2 = \frac{1}{nh^2}S_0(M^{-1} + M^{-1}a_1 a_1^T M^{-1}) , \tag{21}$$

where $M = h^{-2}I_{d\times d} - 2a_2$ assuming $h$ sufficiently small such that $M$ is positive definite. We want to derive $\hat{f}(x) = \exp\{a_0\}$ from the equations. From (20) we get $M^{-1}a_1 = S_1(h/S_0)$. Together with (21), we get $M^{-1} + M^{-1}a_1 a_1^T M^{-1} = S_2(h^2/S_0)$. Hence, $M^{-1} = (S_2/S_0 - (S_1/S_0)(S_1/S_0)^T)h^2 = h^2\Sigma$. Plug them in (19), we obtain the desired expression.

Analogously, for the derivation of the LLDE with degree $p = 1$ in Equation (5), we get

$$\frac{1}{n}S_0 = (2\pi)^{d/2}h^d e^{a_0 + \frac{h^2}{2}a_1^T a_1} , \tag{22}$$

$$\frac{1}{n}S_1 = \frac{h}{n}S_0 a_1 . \tag{23}$$

This gives $a_1 = (1/(hS_0))S_1$, and $e^{a_0} = (S_0/(n(2\pi)^{d/2}h^d)) \exp\{-0.5\|S_1\|^2/S_0^2\}$.

## E  Proof of Lemma 3.1

Let us introduce some notations first. Define $S^{d-1} \equiv \{x \in \mathbb{R}^d : \|x\| = 1\}$ as the unit $(d-1)$-dimensional sphere and $\sigma^{d-1}$ as a normalized spherical measure on $S^{d-1}$. For any $\theta = (\theta_1, \ldots, \theta_m) \in (S^{d-1})^m$ and $x = (x_1, \ldots, x_m) \in \mathbb{R}_+^m$, define $\theta x \equiv (\theta_1 x_1, \ldots, \theta_m x_m) \in \mathbb{R}^{d \times m}$. For any set $B \in \mathbb{R}^{d \times m}$ and $\theta \in (S^{d-1})^m$, define $B_\theta = \{x \in \mathbb{R}_+^m : \theta x \in B\}$. Let $\{\xi_i\}_{i=1}^m$ be i.i.d. random variables uniformly over $S^{d-1}$. Then for any joint random variables $(W_1, \ldots, W_m) \in \mathbb{R}_+^m$ which are independent with $\{\xi_i\}_{i=1}^m$, we have

$$\mathbb{P}\{(\xi_1 W_1, \ldots, \xi_m W_m) \in B\} = \int_{\theta \in (S^{d-1})^m} \mathbb{P}\{(W_1, \ldots, W_m) \in B_\theta \,|\, \theta\} \, d(\sigma^{d-1})^m(\theta) \tag{24}$$

Let $Z = (Z_{1,i}, \ldots, Z_{m,i})$, $\|Z\| = (\|Z_{1,i}\|, \ldots, \|Z_{m,i}\|)$ and let $E = (E_1^{1/d}, \ldots, (\sum_{\ell=1}^m E_\ell)^{1/d})$, then

$$\left| \mathbb{P}\left\{ (c_d n f(x))^{1/d} Z \in B \right\} - \mathbb{P}\left\{ \left( \xi_1 E_1^{1/d}, \ldots, \xi_m(\sum_{\ell=1}^m E_\ell)^{1/d} \right) \in B \right\} \right|$$

$$\leq \left| \mathbb{P}\left\{ (c_d n f(x))^{1/d} Z \in B \right\} - \int_{\theta \in (S^{d-1})^m} \mathbb{P}\{(E_1^{1/d}, \ldots, (\sum_{\ell=1}^m E_\ell)^{1/d}) \in B_\theta \,|\, \theta\} \, d(\sigma^{d-1})^m(\theta) \right|$$

$$\leq \left| \mathbb{P}\left\{ (c_d n f(x))^{1/d} Z \in B \right\} - \int_{\theta \in (S^{d-1})^m} \mathbb{P}\{(c_d n f(x))^{1/d}\|Z\| \in B_\theta \,|\, \theta\} \, d(\sigma^{d-1})^m(\theta) \right|$$

$$+ \int_{\theta \in (S^{d-1})^m} \left| \mathbb{P}\{(c_d n f(x))^{1/d}\|Z\| \in B_\theta \,|\, \theta\} - \mathbb{P}\{E \in B_\theta \,|\, \theta\} \right| d(\sigma^{d-1})^m(\theta) . \tag{25}$$

Now consider the first term in (25). We consider two cases separately.

**Case 1.** If $\|Z_{m,i}\| \geq (\sqrt{n}c_d f(x))^{-1/d}$, we show that the tail events happen with a low probability. Denote $B(x,r) = \{z : \|z - x\| \leq r\}$ and let $p = \mathbb{P}\{t \in B(x, \|Z_{m,i}\|)\} = \int_{B(x, \|Z_{m,i}\|)} f(t)dt$. Since $f$ is twice continuously differentiable, we can see that $p \geq 0.5c_d\|Z_{m,i}\|^d f(x) \geq 0.5/\sqrt{n}$ for sufficiently large $n$. Therefore,

$$\mathbb{P}\{\|Z_{m,i}\| \geq (\sqrt{n}c_d f(x))^{-1/d}\} = \sum_{\ell=0}^{m-1} \binom{n}{\ell} p^\ell (1-p)^{n-\ell} \leq \sum_{\ell=0}^{m-1} n^\ell \left(1 - \frac{1}{2\sqrt{n}}\right)^{(n-\ell)}$$

$$\leq \sum_{\ell=0}^{m-1} n^l e^{-(\sqrt{n}-\ell\sqrt{n})/2} \leq mn^m e^{-(\sqrt{n}-m/\sqrt{n})/2} . \tag{26}$$

**Case 2.** If $\|Z_{m,i}\| < (\sqrt{n}c_d f(x))^{-1/d}$, let $\overline{B} = \{t : (c_d n f(x))^{1/d}t \in B$ and $\|t_m\| < (\sqrt{n}c_d f(x))^{-1/d}\}$ and $\overline{B}_\theta = \{t : (c_d n f(x))^{1/d}t \in B_\theta$ and $t_m < (\sqrt{n}c_d f(x))^{-1/d}\}$. Note that

$$\mathbb{P}(Z \in \widetilde{A}) = (n!/(n-k)!) \int_{t \in \widetilde{A}} \prod_{j=1}^m f(x + t_j) \mathbb{P}_X(|X - x| > |t_m|)^{n-m} dt , \tag{27}$$

which gives

$$\frac{\int_{\theta\in(S^{d-1})^m}\mathbb{P}\{(c_dnf(x))^{1/d}\|Z\|\in B_\theta, \|Z_{m,i}\|<(\sqrt{n}c_df(x))^{-1/d}\,|\,\theta\}\,d(\sigma^{d-1})^m(\theta)}{\mathbb{P}\{(c_dnf(x))^{1/d}Z\in B, \|Z_{m,i}\|<(\sqrt{n}c_df(x))^{-1/d}\}}$$

$$=\frac{\int_{\theta\in(S^{d-1})^m}\mathbb{P}\{\|Z\|\in\overline{B_\theta}\,|\,\theta\}\,d(\sigma^{d-1})^m(\theta)}{\mathbb{P}\{Z\in\overline{B}\}}$$

$$=\frac{\int_{\theta\in(S^{d-1})^m}\frac{n!}{(n-k)!}\left(\int_{t\in\overline{B_\theta}}\left(\prod_{j=1}^m f(x+\theta_jt_j)\right)(\mathbb{P}\{\|X-x\|>\|t_m\|\})^{n-m}\,dt\right)d(\sigma^{d-1})^m(\theta)}{\frac{n!}{(n-k)!}\int_{t\in\overline{B}}\left(\prod_{j=1}^m f(x+t_j)\right)(\mathbb{P}\{\|X-x\|>\|t_m\|\})^{n-m}\,dt}$$

$$\leq\frac{\sup_{\theta\in(S^{d-1})^m}\sup_{t\in\overline{B_\theta}}\prod_{j=1}^m f(x+\theta_jt_j)}{\inf_{t\in\overline{B}}\prod_{j=1}^m f(x+t_j)}$$

$$\leq\left(\frac{\sup_{\|t\|\leq(\sqrt{n}c_df(x))^{-1/d}}f(x+t)}{\inf_{\|t\|\leq(\sqrt{n}c_df(x))^{-1/d}}f(x+t)}\right)^m, \tag{28}$$

where the first inequality follows from the fact that $\int_{\theta\in(S^{d-1})^m}(\int_{\overline{B_\theta}}g(t_m)dt)d(\sigma^{d-1})^m(\theta)=\int_{\overline{B}}g(\|t_m\|)dt$. Since $f$ is continuously differentiable, by mean value theorem, there exists $a,b\in B(x,(\sqrt{n}c_df(x))^{-1/d})$ such that

$$\frac{\sup_{\|t\|\leq(\sqrt{n}c_df(x))^{-1/d}}f(x+t)}{\inf_{\|t\|\leq(\sqrt{n}c_df(x))^{-1/d}}f(x+t)}=\frac{f(b)+(a-b)^T\nabla f(a)}{f(b)}\leq 1+\frac{2(\sqrt{n}c_df(x))^{-1/d}\|\nabla f(a)\|}{f(b)} \tag{29}$$

By the assumption, there exists a ball $B(x,\varepsilon)$ such that $\|\nabla f(a)\|=O(1)$ and $f(a)>0$ for all $a\in B(x,\varepsilon)$, so for sufficiently large $n$ such that $(\sqrt{n}c_df(x))^{-1/d}<\varepsilon$, there exists some constant $C$ such that $\sup_{\|t\|\leq(\sqrt{n}c_df(x))^{-1/d}}f(x+t)\leq(1+Cn^{-1/(2d)})\inf_{\|t\|\leq(\sqrt{n}c_df(x))^{-1/d}}f(x+t)$. Therefore, (28) is upper bounded by $(1+Cn^{-1/(2d)})^m$. Similarly, (28) is lower bounded by $(1-Cn^{-1/(2d)})^m$.

For simplicity, let $\mathcal{E}=\{\|Z_{m,i}\|<(\sqrt{n}c_df(x))^{-1/d}\}$. Then combining the two cases, the first term in (25) is bounded by:

$$\left|\mathbb{P}\left\{(c_dnf(x))^{1/d}Z\in B\right\}-\int_{\theta\in(S^{d-1})^m}\mathbb{P}\{(c_dnf(x))^{1/d}\|Z\|\in B_\theta\,|\,\theta\}\,d(\sigma^{d-1})^m(\theta)\right|$$

$$\leq\mathbb{P}\left\{(c_dnf(x))^{1/d}Z\in B,\mathcal{E}^C\right\}+\int_{\theta\in(S^{d-1})^m}\mathbb{P}\{(c_dnf(x))^{1/d}\|Z\|\in B_\theta,\mathcal{E}^C\,|\,\theta\}\,d(\sigma^{d-1})^m(\theta)$$

$$+\left|\mathbb{P}\left\{(c_dnf(x))^{1/d}Z\in B,\mathcal{E}\right\}-\int_{\theta\in(S^{d-1})^m}\mathbb{P}\{(c_dnf(x))^{1/d}\|Z\|\in B_\theta,\mathcal{E}\,|\,\theta\}\,d(\sigma^{d-1})^m(\theta)\right|$$

$$\leq\mathbb{P}\{\mathcal{E}^C\}+\int_{\theta\in(S^{d-1})^m}\mathbb{P}\{\mathcal{E}^C\}\,d(\sigma^{d-1})^m(\theta)$$

$$+\mathbb{P}\left\{(c_dnf(x))^{1/d}Z\in B,\mathcal{E}\right\}\left|1-\frac{\int_{\theta\in(S^{d-1})^m}\mathbb{P}\{(c_dnf(x))^{1/d}\|Z\|\in B_\theta,\mathcal{E}\,|\,\theta\}\,d(\sigma^{d-1})^m(\theta)}{\mathbb{P}\left\{(c_dnf(x))^{1/d}Z\in B,\mathcal{E}\right\}}\right|$$

$$\leq 2\mathbb{P}\{\mathcal{E}^C\}+\mathbb{P}\left\{(c_dnf(x))^{1/d}Z\in B,\mathcal{E}\right\}\max\{(1+Cn^{-1/(2d)})^m-1,1-(1-Cn^{-1/(2d)})^m\}$$

$$\leq 2mn^me^{-(\sqrt{n}-m/\sqrt{n})/2}+\max\{(1+Cn^{-1/(2d)})^m-1,1-(1-Cn^{-1/(2d)})^m\}. \tag{30}$$

Now consider the second term of (25). We will use Corollary 5.5.5 of [29] to show that this term vanishes for $m=O(\log n)$ and as $n$ grows.

**Lemma E.1** (Corollary 5.5.5, [29]). *Let $Y_1,Y_2,\ldots,Y_n$ be i.i.d. samples from unknown distribution with pdf $f$. Let $Y_{1:n}\leq Y_{2:n}\leq\cdots\leq Y_{n:n}$ be the order statistics. Assume the density $f$ satisfies $|\log f(y)|\leq Ly^\delta$ for $0<y<y_0$ and $f(y)=0$ for $y<0$, where $L$ and $\delta$ are constants. Then*

$$d_{\mathrm{TV}}\left(n\,(Y_{1:n},Y_{2:n},\ldots,Y_{m:n}),\left(E_1,E_1+E_2,\ldots,\sum_{j=1}^m E_j\right)\right)\leq C_0\left((m/n)^\delta m^{1/2}+m/n\right), \tag{31}$$

*where $C_0 > 0$ is a constant. $E_1, \ldots, E_m$ are i.i.d standard exponential random variables.*

Now for fixed $x$, consider the distribution of $c_d f(x)\|X - x\|^d$ denoted by $\tilde{P}$. Define $Y_1, Y_2, \ldots, Y_n$ drawn i.i.d. from $\tilde{P}$. We can see that $c_d f(x)\|Z\|^d \overset{\mathcal{L}}{=} (Y_{1:n}, \ldots, Y_{m:n})$, where $\overset{\mathcal{L}}{=}$ denotes equivalence in distribution. The pdf $\tilde{f}$ of $\tilde{P}$ is given by:

$$\tilde{f}(t) = \frac{d}{dt}\mathbb{P}\{c_d f(x)\|X - x\|^d \leq t\} = \frac{d}{dt}\int_{y \in B(x, r_t)} f(y)dy . \tag{32}$$

where $r_t = (t/(c_d f(x)))^{1/d}$. Here we have:

$$\frac{dr_t}{dt} = \frac{t^{1/d - 1}(c_d f(x))^{-1/d}}{d} = \frac{1}{f(x)dc_d r_t^{d-1}} . \tag{33}$$

If $f$ is twice continuously differentiable, we have:

$$
\begin{aligned}
\left| \tilde{f}(t) - 1 \right| &= \left| \frac{d}{dt}\int_{y \in B(x, r_t)} f(y)dy - 1 \right| = \left| \frac{dr_t}{dt}\left(\frac{d}{dr_t}\int_{y \in B(x, r_t)} f(y)dy\right) - 1 \right| \\
&= \frac{1}{f(x)dc_d r_t^{d-1}}\left| \frac{d}{dr_t}\left(\int_{y \in B(x, r_t)} f(y)dy\right) - f(x)dc_d r_t^{d-1} \right| \\
&= \frac{1}{f(x)dc_d r_t^{d-1}}\left| \int_{y \in S^{d-1}(x, r_t)} (f(y) - f(x))d\sigma^{d-1}(y) \right| ,
\end{aligned}
\tag{34}
$$

where $S^{d-1}$ is the $(d - 1)$-sphere centered at $x$ with radius $r_t$ and $\sigma^{d-1}$ is the spherical measure. By mean value theorem, there exists $a(y) \in B(x, r_t)$ such that $f(y) - f(x) = (y - x)^T \nabla f(x) + (a(y) - x)^T H_f(a(y))(a(y) - x)$, where $a(y)$ depends on $y$. Therefore,

$$
\begin{aligned}
&\left| \int_{y \in S^{d-1}(x, r_t)} (f(y) - f(x))d\sigma^{d-1}(y) \right| \\
&= \left| \underbrace{\int_{y \in S^{d-1}(x, r_t)} (y - x)^T \nabla f(x)d\sigma^{d-1}(y)}_{=0} + \int_{y \in S^{d-1}(x, r_t)} (a(y) - x)^T H_f(a(y))(a(y) - x)d\sigma^{d-1}(y) \right| \\
&\leq \left( \sup_{a \in B(x, r_t)} \|H_f(a)\| \|a - x\|^2 \right) \sigma^{d-1}(S^{d-1}(x, r_t)) \\
&\leq dc_d r_t^{d+1}\left( \sup_{a \in B(x, r_t)} \|H_f(a)\| \right)
\end{aligned}
\tag{35}
$$

Since there exists a ball $B(x, \varepsilon)$ such that $\|H_f(a)\| = O(1)$ for all $a \in B(x, \varepsilon)$. Therefore, for sufficiently small $t$ such that $r_t < \varepsilon$, we have:

$$\left| \tilde{f}(t) - 1 \right| \leq \frac{dc_d r_t^{d+1}\left( \sup_{a \in B(x, r_t)} \|H_f(a)\| \right)}{f(x)dc_d r_t^{d-1}} = \frac{r_t^2\left( \sup_{a \in B(x, r_t)} \|H_f(a)\| \right)}{f(x)} . \tag{36}$$

Recall that $r_t = (t/(c_d f(x)))^{1/d}$, so there exists $L > 0$ such that $|\tilde{f}(t) - 1| \leq Lt^{2/d}$ for sufficiently small $t$. Hence, $|\log \tilde{f}(t)| \leq L't^{2/d}$ for some $L' > 0$ and sufficiently small $t$. So $\tilde{f}$ satisfies the condition in Lemma. E.1 with $\delta = 2/d$. Therefore, for any $B_\theta \subseteq \mathbb{R}_+^m$, we have:

$$
\begin{aligned}
&\left| \mathbb{P}\{(c_d n f(x))^{1/d}\|Z\| \in B_\theta\} - \mathbb{P}\{E \in B_\theta\} \right| \\
&\leq d_{\mathrm{TV}}\left( c_d n f(x)\|Z\|^d, \left( E_1, E_1 + E_2, \ldots, \sum_{j=1}^m E_j \right) \right) \\
&\leq C_0\left( (\frac{m}{n})^{2/d}m^{1/2} + \frac{m}{n} \right) .
\end{aligned}
\tag{37}
$$

Therefore, by combing (30) and (37), we have:

$$\left| \mathbb{P}\left\{ (c_d n f(x))^{1/d} Z \in B \right\} - \mathbb{P}\left\{ \left( \xi_1 E_1^{1/d}, \ldots, \xi_m (\sum_{l=1}^{m} E_\ell)^{1/d} \right) \in B \right\} \right|$$

$$\leq 2mn^m e^{-\frac{\sqrt{n} - m/\sqrt{n}}{2}} + \max\{(1 + Cn^{\frac{-1}{2d}})^m - 1, 1 - (1 - Cn^{\frac{-1}{2d}})^m\} + C_0 \left( (\frac{m}{n})^{\frac{2}{d}} m^{\frac{1}{2}} + \frac{m}{n} \right) \quad (38)$$

for any set $B \in \mathbb{R}^{d \times m}$. Therefore, the total variation distance $d_{\text{TV}}((c_d n f(x))^{1/d}(Z_{1,i}, Z_{2,i}, \ldots, Z_{m,i}), (\xi_1 E_1^{1/d}, \xi_2 (E_1 + E_2)^{1/d}, \ldots, \xi_m (\sum_{\ell=1}^{m} E_\ell)^{1/d}))$ is bounded by the RHS quantity. By taking $m = O(\log n)$, the RHS converges to 0 as $n$ goes to infinity. Therefore, we have the desired statement.

## F  Proof of Theorem 1

We first compute the asymptotic bias. We define new notations to represent the estimate as

$$\widehat{H}_k^{(n)} = \frac{1}{n} \sum_{i=1}^{n} \left\{ \underbrace{h\big( (c_d n f(X_i))^{1/d} Z_{k,i}, S_{0,i}, S_{1,i}, S_{2,i} \big) - \log f(X_i)}_{\equiv H_i} \right\} ,$$

where $h : \mathbb{R}^d \times \mathbb{R} \times \mathbb{R}^d \times \mathbb{R}^{d \times d} \to \mathbb{R}$ is defined as

$$h(t_1, t_2, t_3, t_4) =$$

$$d \log \|t_1\| + d \log(2\pi) - \log c_d - \log t_2 + \frac{1}{2} \log \left( \det \left( \frac{t_4}{t_2} - \frac{t_3 t_3^T}{t_2^2} \right) \right) + \frac{1}{2} t_3^T (t_4 - t_3 t_3^T)^{-1} t_3 . \quad (39)$$

Let $H_i \equiv h((c_d n f(X_i))^{1/d} Z_{k,i}, S_{0,i}, S_{1,i}, S_{2,i})) - \log f(X_i)$. Since the terms $H_1, H_2, \ldots, H_n$ are identically distributed, the expected value of $\widehat{H}_k^{(n)}$ converges to

$$\lim_{n \to \infty} \mathbb{E}[\widehat{H}_k^{(n)}] = \lim_{n \to \infty} \mathbb{E}[H_1] = \lim_{n \to \infty} \mathbb{E}_{X_1} \big[ \mathbb{E}[H_1 | X_1] \big] \quad (40)$$

Typical approach of dominated convergence theorem cannot be applied to the above limit, since analyzing $\mathbb{E}[H_1 | X_1]$ for finite sample $n$ is challenging. In order to exchange the limit with the (conditional) expectation, we assume the following Ansatz 1 to be true. As noted in [28] this is common in the literature on consistency of $k$-NN estimators, where the same assumptions have been implicitly made without explicitly stating as such, in existing analyses of entropy estimators including [15, 9, 18, 39]. This assumption can be avoided for Renyi entropy as in the proof of consistency in [28] or for sharper results such as the convergence rate of the bias with respect to the sample size but with more assumptions as in [8, 33, 1].

**Ansatz 1.** *The following exchange of limit holds:*

$$\lim_{n \to \infty} \mathbb{E}[H_1] = \mathbb{E}_{X_1} \left[ \lim_{n \to \infty} \mathbb{E}[H_1 | X_1] \right] , \quad (41)$$

Under this ansatz, perhaps surprisingly, we will show that the expectation inside converges to $-\log f(X_1)$ plus some bias that is independent of the underlying distribution. Precisely, for almost every $x$ and given $X_1 = x$,

$$\mathbb{E}[H_1 | X_1 = x] + \log f(x) = \mathbb{E}\left[ h((c_d n f(x))^{1/d} Z_{k,i}, S_{0,1}, S_{1,i}, S_{2,i}) \right]$$
$$\longrightarrow B_{k,d} , \quad (42)$$

as $n \to \infty$ where $B_{k,d}$ is a constant that only depends on $k$ and $d$, defined in (44). This implies that

$$\mathbb{E}_{X_1} \left[ \lim_{n \to \infty} \mathbb{E}[H_1 | X_1] \right] = \mathbb{E}_{X_1}[-\log f(X_1) + B_{k,d}]$$
$$= H(X) + B_{k,d} . \quad (43)$$

Together with (40), this finishes the proof of the desired claim.

We are now left to prove the convergence of (42). We first give a formal definition of the bias $B_{k,d}$ by replacing the sample defined quantities by a similar quantities defined from order-statistics, and use Lemma 3.1 to prove the convergence. Recall that our order-statistics is defined by two sequences of $m$ i.i.d. random variables: i.i.d. standard exponential random variables $E_1, \ldots, E_m$ and i.i.d. random variables $\xi_1, \ldots, \xi_m$ uniformly distributed over $S^{d-1}$. We define

$$B_{k,d} \equiv \mathbb{E}\left[ h\left( \xi_k \left( \sum_{\ell=1}^{k} E_\ell \right)^{1/d}, \tilde{S}_0^{(\infty)}, \tilde{S}_1^{(\infty)}, \tilde{S}_2^{(\infty)} \right) \right], \tag{44}$$

where, as we will show, $\tilde{S}_\alpha^{(\infty)}$ is the limit of empirical quantity $S_{\alpha,i}$ defined from samples for each $\alpha \in \{0, 1, 2\}$, and we know that $(c_d n f(x))^{1/d} Z_{k,i}$ converges to $\xi_k (\sum_{\ell=1}^{k} E_\ell)^{1/d}$ for almost every $x$ from Lemma 3.1. $S^{(\infty)}$ is defined by a convergent random sequence.

$$\tilde{S}_\alpha^{(m)} \equiv \sum_{j=1}^{m} \frac{\xi_j^{(\alpha)}(\sum_{\ell=1}^{j} E_\ell)^{\alpha/d}}{(\sum_{\ell=1}^{k} E_\ell)^{\alpha/d}} \exp\left\{ -\frac{(\sum_{\ell=1}^{j} E_\ell)^{2/d}}{2(\sum_{\ell=1}^{k} E_\ell)^{2/d}} \right\}, \tag{45}$$

where $\xi_j^{(0)} = 1$, $\xi_j^{(1)} = \xi_j$, $\xi_j^{(2)} = \xi_j \xi_j^T$ and $\tilde{S}_\alpha^{(\infty)} = \lim_{m \to \infty} \tilde{S}_\alpha^{(m)}$. This limit exists, since $\tilde{S}_0^{(m)}$ is non-decreasing in $m$, and the convergence of $\tilde{S}_1^{(m)}$ and $\tilde{S}_2^{(m)}$ follows from Lemma F.1. We introduce simpler notations for the joint random variables: $\tilde{S}^{(m)} = (\xi_k(\sum_{\ell=1}^{k} E_\ell)^{1/d}, \tilde{S}_0^{(m)}, \tilde{S}_1^{(m)}, \tilde{S}_2^{(m)})$ and $\tilde{S}^{(\infty)} = (\xi_k(\sum_{\ell=1}^{k} E_\ell)^{1/d}, \tilde{S}_0^{(\infty)}, \tilde{S}_1^{(\infty)}, \tilde{S}_2^{(\infty)})$. Considering the quantities $S^{(n)} = ((c_d n f(x))^{1/d} Z_{k,i}, S_{0,i}, S_{1,i}, S_{2,i})$ defined from samples, we show that this converges to $\tilde{S}^{(\infty)}$. Precisely, applying triangular inequality,

$$d_{\mathrm{TV}}(S^{(n)}, \tilde{S}^{(\infty)}) \leq d_{\mathrm{TV}}(S^{(n)}, \tilde{S}^{(m)}) + d_{\mathrm{TV}}(\tilde{S}^{(m)}, \tilde{S}^{(\infty)}), \tag{46}$$

and we show that both terms converge to zero for any $m = \Theta(\log n)$. Given that $h$ is continuous and bounded, this implies that

$$\lim_{n \to \infty} \mathbb{E}[H_1 | X_1 = x] = \mathbb{E}[\lim_{n \to \infty} h(S^{(n)}) - \log f(x) | X_1 = x]$$
$$= -\log f(x) + \mathbb{E}[h(\tilde{S}^{(\infty)})],$$

for almost every $x$, proving (43).

The convergence of the first term follows from Lemma 3.1. Precisely, consider the function $g_m : \mathbb{R}^{d \times m} \to \mathbb{R}^d \times \mathbb{R} \times \mathbb{R}^d \times \mathbb{R}^{d \times d}$ defined as:

$$g_m(t_1, t_2, \ldots, t_m) = \left( t_k, \sum_{j=1}^{m} \exp\{-\frac{\|t_j\|^2}{2\|t_k\|^2}\}, \sum_{j=1}^{m} \frac{t_j}{\|t_k\|} \exp\{-\frac{\|t_j\|^2}{2\|t_k\|^2}\}, \sum_{j=1}^{m} \frac{t_j t_j^T}{\|t_k\|^2} \exp\{-\frac{\|t_j\|^2}{2\|t_k\|^2}\} \right), \tag{47}$$

such that $S^{(n)} = g_m\left( (c_d n f(x))^{1/d} (Z_{1,i}, Z_{2,i}, \ldots, Z_{m,i}) \right)$, which follows from the definition of $S^{(n)} = ((c_d n f(x))^{1/d} Z_{k,i}, S_{0,i}, S_{1,i}, S_{2,i})$ in (10). Similarly, $\tilde{S}^{(m)} = g_m\left( \xi_1 E_1^{1/d}, \xi_2 (E_1 + E_2)^{1/d}, \ldots \xi_m (\sum_{\ell=1}^{m} E_\ell)^{1/d} \right)$. Since $g_m$ is continuous, so for any set $A \in \mathbb{R}^d \times \mathbb{R} \times \mathbb{R}^d \times \mathbb{R}^{d \times d}$, there exists a set $\widetilde{A} \in \mathbb{R}^{d \times m}$ such that $g_m(\widetilde{A}) = A$. So for any $x$ such that there exists $\varepsilon > 0$ such that $f(a) > 0$, $\|\nabla f(a)\| = O(1)$ and $\|H_f(a)\| = O(1)$ for any $\|a - x\| < \varepsilon$, we have:

$d_{\mathrm{TV}}(S^{(n)}, \tilde{S}^{(m)})$

$$= \sup_{A} \left| \mathbb{P}\left\{ g_m\left( (c_d n f(x))^{\frac{1}{d}} Z_{1,i}, \ldots, (c_d n f(x))^{\frac{1}{d}} Z_{m,i} \right) \in A \right\} - \mathbb{P}\{ g_m(\xi_1 E_1^{\frac{1}{d}}, \ldots \xi_m(\sum_{l=1}^{m} E_\ell)^{\frac{1}{d}}) \in A \} \right|$$

$$\leq \sup_{\widetilde{A} \in \mathbb{R}^{d \times m}} \left| \mathbb{P}\left\{ \left( (c_d n f(x))^{1/d} Z_{1,i}, \ldots, (c_d n f(x))^{1/d} Z_{m,i} \right) \in \widetilde{A} \right\} - \mathbb{P}\{ (\xi_1 E_1^{1/d}, \ldots \xi_m(\sum_{\ell=1}^{m} E_\ell)^{1/d}) \in \widetilde{A} \} \right|$$

$$= d_{\mathrm{TV}}\left( \left( (c_d n f(x))^{1/d} Z_{1,i}, \ldots, (c_d n f(x))^{1/d} Z_{m,i} \right), \left( \xi_1 E_1^{1/d}, \ldots \xi_m(\sum_{\ell=1}^{m} E_\ell)^{1/d} \right) \right)$$

$$\xrightarrow{n \to \infty} 0, \tag{48}$$

where the last inequality follows from Lemma 3.1. By the assumption that $f$ has open support and $\|\nabla f\|$ and $\|H_f\|$ is bounded almost everywhere, this convergence holds for almost every $x$.

For the second term in (46), let $\tilde{T}_\alpha^{(m)} = \tilde{S}_\alpha^{(\infty)} - \tilde{S}_\alpha^{(m)}$ and we claim that $\tilde{T}_\alpha^{(m)}$ converges to 0 in distribution by the following lemma.

**Lemma F.1.** *Assume $m \to \infty$ as $n \to \infty$ and $k \geq 3$, then*

$$\lim_{n\to\infty} \mathbb{E}\| \tilde{T}_\alpha^{(m)} \| = 0 \tag{49}$$

*for any $\alpha \in \{0,1,2\}$. Hence $(\tilde{T}_0^{(m)}, \tilde{T}_1^{(m)}, \tilde{T}_2^{(m)})$ converges to $(0,0,0)$ in distribution.*

This implies that $(\tilde{S}_0^{(m)}, \tilde{S}_1^{(m)}, \tilde{S}_2^{(m)})$ converges to $(\tilde{S}_0^{(\infty)}, \tilde{S}_1^{(\infty)}, \tilde{S}_2^{(\infty)})$ in distribution, i.e.,

$$d_{\text{TV}}(\tilde{S}^{(m)}, \tilde{S}^{(\infty)}) \overset{n\to\infty}{\longrightarrow} 0 \,, \tag{50}$$

Combine (48) and (50) in (46), this implies the desired claim.

We next prove the upper bound on the variance, following the technique from [2, Section 7.3]. For the usage of Efron-Stein inequality, we need a second set of i.i.d. samples $\{X_1', X_2', \ldots, X_n'\}$. For simplicity, denote $\widehat{H} = \widehat{H}_{kLNN}^{(n)}(X)$ be the kLNN estimate base on original sample $\{X_1, \ldots, X_n\}$ and $\widehat{H}^{(i)}$ be the kLNN estimate based on $\{X_1, \ldots, X_{i-1}, X_i', X_{i+1}, \ldots X_n\}$. Then Efron-Stein theorem states that

$$\text{Var}\left[\widehat{H}\right] \leq 2\sum_{j=1}^n \mathbb{E}\left[ \left(\widehat{H} - \widehat{H}^{(j)}\right)^2 \right] \,. \tag{51}$$

Recall that

$$\widehat{H} = \frac{1}{n}\sum_{i=1}^n \Big\{ \underbrace{h\big( (c_d n f(X_i))^{1/d} Z_{k,i}, S_{0,i}, S_{1,i}, S_{2,i} \big) - \log f(X_i)}_{\equiv H_i} \Big\} \,,$$

where $h : \mathbb{R}^d \times \mathbb{R} \times \mathbb{R}^d \times \mathbb{R}^{d\times d} \to \mathbb{R}$ is defined as

$$h(t_1, t_2, t_3, t_4) =$$
$$d\log\|t_1\| + d\log(2\pi) - \log c_d - \log t_2 + \frac{1}{2}\log\left( \det\left( \frac{t_4}{t_2} - \frac{t_3 t_3^T}{t_2^2} \right) \right) + \frac{1}{2} t_3^T (t_4 - t_3 t_3^T)^{-1} t_3 \,. \tag{52}$$

Similarly, we can write $\widehat{H}^{(j)} = \frac{1}{n}\sum_{i=1}^n H_i^{(j)}$ for any $j \in \{1, \ldots, n\}$. Therefore, the difference of $\widehat{H}$ and $\widehat{H}^{(j)}$ can be bounded by:

$$\widehat{H} - \widehat{H}^{(j)} = \frac{1}{n}\sum_{i=1}^n \left( H_i - H_i^{(j)} \right) \,. \tag{53}$$

Notice that $H_i$ only depends on $X_i$ and its $m$ nearest neighbors, so $H_i - H_i^{(j)} = 0$ if none of $X_j$ and $X_j'$ are in $m$ nearest neighbor of $X_i$. If we denote $Z_{i,j} = \mathbb{I}\{X_j$ is in $m$ nearest neighbor of $X_i\}$, then $H_i = H_i^{(j)}$ if $Z_{i,j} + Z_{i,j'} = 0$. According to [2, Lemma 20.6], since $X$ has a density, with probability one, $\sum_{i=1}^n Z_{i,j} \leq m\gamma_d$, where $\gamma_d$ is the minimal number of cones of angle $\pi/6$ that can cover $\mathbb{R}^d$, which only depends on $d$. Similarly, $\sum_{i=1}^n Z_{i,j'} \leq m\gamma_d$. If we denote $S = \{i : Z_{i,j} + Z_{i,j'} > 0\}$, the cardinality of $S$ satisfy $|S| \leq 2m\gamma_d$. Therefore, we have $\widehat{H} - \widehat{H}^{(j)} = \frac{1}{n}\sum_{i\in S}\left( H_i - H_i^{(j)} \right)$.

By Cauchy-Schwarz inequality, we have

$$
\begin{aligned}
\mathbb{E}\left[\left(\widehat{H} - \widehat{H}^{(j)}\right)^2\right] &= \mathbb{E}\left[\frac{1}{n^2}\left(\sum_{i \in S}\left(H_i - H_i^{(j)}\right)\right)^2\right] \\
&\leq \mathbb{E}\left[\frac{|S|}{n^2}\sum_{i \in S}\left(H_i - H_i^{(j)}\right)^2\right] \\
&= \frac{|S|}{n^2}\sum_{i \in S}\mathbb{E}\left[\left(H_i - H_i^{(j)}\right)^2\right] \\
&\leq \frac{2|S|}{n^2}\sum_{i \in S}\left(\mathbb{E}\left[H_i^2\right] + \mathbb{E}\left[(H_i^{(j)})^2\right]\right) .
\end{aligned} \tag{54}
$$

Notice that $H_i$'s and $H_i^{(j)}$'s are identically distributed, so we are left to compute $\mathbb{E}\left[H_1^2\right]$. Conditioning on $X_1 = x$, similarly to (42), we have

$$
\begin{aligned}
\mathbb{E}\left[(H_1 + \log f(x))^2 | X_1 = x\right] &= \mathbb{E}\left[h^2((c_d n f(x))^{1/d} Z_{k,i}, S_{0,1}, S_{1,i}, S_{2,i})\right] \\
&\longrightarrow B_{k,d}^{(2)},
\end{aligned} \tag{55}
$$

as $n \to \infty$, where $B_{k,d}^{(2)} \equiv \mathbb{E}\left[h^2\left(\xi_k\left(\sum_{\ell=1}^k E_\ell\right)^{1/d}, \tilde{S}_0^{(\infty)}, \tilde{S}_1^{(\infty)}, \tilde{S}_2^{(\infty)}\right)\right]$. Therefore,

$$
\begin{aligned}
\mathbb{E}\left[H_1^2 | X_1 = x\right] &= B_{k,d}^{(2)} - 2\log f(x)\mathbb{E}\left[H_1 | X_1 = x\right] - (\log f(x))^2 \\
&= B_{k,d}^{(2)} - 2\log f(x)B_{k,d} + (\log f(x))^2 .
\end{aligned} \tag{56}
$$

Take expectation over $X_1$, we obtain:

$$
\begin{aligned}
\mathbb{E}[H_1^2] &= \mathbb{E}_{X_1}\left[\lim_{n \to \infty}\mathbb{E}\left[H_1^2 | X_1\right]\right] = \mathbb{E}_{X_1}\left[B_{k,d}^{(2)} - 2\log f(X_1)B_{k,d} + (\log f(X_1))^2\right] \\
&= B_{k,d}^{(2)} + 2H(X)B_{k,d} + \int f(x)(\log f(x))^2 dx < +\infty ,
\end{aligned} \tag{57}
$$

where the last inequality comes from the assumption that $\int f(x)(\log f(x))^2 dx < +\infty$. Combining with (51) and (54), we have

$$
\operatorname{Var}\left[\widehat{H}\right] \leq 2\sum_{j=1}^n \mathbb{E}\left[\left(\widehat{H} - \widehat{H}^{(j)}\right)^2\right] \leq \frac{4|S|}{n}\sum_{i \in S}\left(\mathbb{E}\left[H_i^2\right] + \mathbb{E}\left[(H_i^{(j)})^2\right]\right) \leq \frac{8|S|^2 C_2}{n} \leq \frac{32m^2\gamma_d^2 C_2}{n} , \tag{58}
$$

where $C_2$ is the upper bound for $\mathbb{E}[H_1^2]$. Take $m = O(\log n)$ then the proof is complete.

### F.1   Proof of Lemma F.1

Firstly, since $|\xi_i| = 1$, we can upper bound the expectation of $\mathbb{E}\|\tilde{T}_{\alpha,i}^{(m)}\|$ by:

$$
\begin{aligned}
\mathbb{E}\|\tilde{T}_{\alpha,i}^{(m)}\| &= \mathbb{E}\left\|\sum_{j=m+1}^\infty \frac{\xi_j^{(\alpha)}(\sum_{\ell=1}^j E_\ell)^{\alpha/d}}{(\sum_{\ell=1}^k E_\ell)^{\alpha/d}}\exp\left\{-\frac{(\sum_{\ell=1}^j E_\ell)^{2/d}}{2(\sum_{\ell=1}^k E_\ell)^{2/d}}\right\}\right\| \\
&\leq \sum_{j=m+1}^\infty \mathbb{E}\left\|\frac{\xi_j^{(\alpha)}(\sum_{\ell=1}^j E_\ell)^{\alpha/d}}{(\sum_{\ell=1}^k E_\ell)^{\alpha/d}}\exp\left\{-\frac{(\sum_{\ell=1}^j E_\ell)^{2/d}}{2(\sum_{\ell=1}^k E_\ell)^{2/d}}\right\}\right\| \\
&= \sum_{j=m+1}^\infty \mathbb{E}\left|\frac{(\sum_{\ell=1}^j E_\ell)^{\alpha/d}}{(\sum_{\ell=1}^k E_\ell)^{\alpha/d}}\exp\left\{-\frac{(\sum_{\ell=1}^j E_\ell)^{2/d}}{2(\sum_{\ell=1}^k E_\ell)^{2/d}}\right\}\right| .
\end{aligned} \tag{59}
$$

Notice that the expression is a function of $(\sum_{\ell=1}^j E_\ell / \sum_{\ell=1}^k E_\ell)^{1/d} \equiv R_j$ for $j > m$, we will identify the distribution of $R_j$ first. For any fixed $j \geq k$, let $T_k = \sum_{\ell=1}^k E_\ell$ and $T_{j-k} = \sum_{\ell=k+1}^j E_\ell$, such

that $R_j = ((T_k + T_{j-k})/T_k)^{1/d}$. Notice that $T_k$ is the summation of $k$ i.i.d. standard exponential random variables, so $T_k \sim$ *Erlang* $(k, 1)$. Similarly, $T_{j-k} \sim$ *Erlang* $(j-k, 1)$. Also $T_k$ and $T_{j-k}$ are independent. Recall that the pdf of *Erlang* $(k, \lambda)$ is given by $f_{k,\lambda}(x) = \lambda^k x^{k-1} e^{-\lambda x}/(k-1)!$ for $x \geq 0$. Therefore, the CDF of $R_j$ is given by:

$$
\begin{aligned}
F_{R_j}(t) &= \mathbb{P}\{R_j \leq t\} = \mathbb{P}\{(\frac{T_k + T_{j-k}}{T_k})^{1/d} \leq t\} = \mathbb{P}\{\frac{T_{j-k}}{T_k} \leq t^d - 1\} \\
&= \int_{x \geq 0} \frac{x^{k-1}e^{-x}}{(k-1)!} \left( \int_{y=0}^{(t^d-1)x} \frac{y^{j-k-1}e^{-y}}{(j-k-1)!}dy \right) dx \\
&= \int_{x \geq 0} \frac{x^{k-1}e^{-x}}{(k-1)!} \left( 1 - \sum_{\ell=0}^{j-k-1} \frac{1}{\ell!}x^\ell(t^d-1)^\ell e^{-x(t^d-1)} \right) dx \\
&= 1 - \sum_{\ell=0}^{j-k-1} \left( \int_{x \geq 0} \frac{x^{k-1}e^{-x}}{(k-1)!} \frac{1}{\ell!}x^\ell(t^d-1)^\ell e^{-x(t^d-1)}dx \right) \\
&= 1 - \sum_{\ell=0}^{j-k-1} \left( \frac{(t^d-1)^\ell}{(k-1)!\ell!} \int_{x \geq 0} x^{k-1+\ell}e^{-xt^d}dx \right) \\
&= 1 - \sum_{\ell=0}^{j-k-1} \frac{(t^d-1)^\ell}{(k-1)!\ell!} (k-1+\ell)! \, t^{-d(k-1+\ell)} \\
&= 1 - \sum_{\ell=0}^{j-k-1} \binom{k-1+\ell}{\ell} t^{-d(k-1)}(1-t^{-d})^\ell ,
\end{aligned}
\tag{60}
$$

for $t \in [1, +\infty)$. Given the CDF of $R_j$, each term in (66) is upper bounded by:

$$
\begin{aligned}
& \mathbb{E} \left| \frac{(\sum_{\ell=1}^j E_\ell)^{\alpha/d}}{(\sum_{\ell=1}^k E_\ell)^{\alpha/d}} \exp\{-\frac{(\sum_{\ell=1}^j E_\ell)^{2/d}}{2(\sum_{\ell=1}^k E_\ell)^{2/d}}\} \right| = \mathbb{E}_{R_j} \left| t^\alpha e^{-t^2} \right| \leq \mathbb{E}_{R_j} \left[ t^2 e^{-t^2} \right] \\
&= \int_{t=1}^\infty t^2 e^{-t^2} dF_{R_j}(t) = t^2 e^{-t^2} F_{R_j}(t) \Big|_1^\infty - \int_{t=1}^\infty F_{R_j}(t)d(t^2 e^{-t^2}) \\
&= -\int_{t=1}^\infty (2te^{-t^2} - 2t^3 e^{-t^2})F_{R_j}(t)dt = \int_{t=1}^\infty 2t(t^2-1)e^{-t^2}F_{R_j}(t)dt .
\end{aligned}
\tag{61}
$$

Therefore, in order to establish an upper bound for (66), we need an upper bound for $F_{R_j}(t)$. Here we will consider two cases depending on $t$. If $t > (j/2k)^{1/d}$, we just use the trivial upper bound $F_{R_j}(t) < 1$. If $1 \leq t \leq (j/2k)^{1/d}$, since $t^d \geq 1$, we have:

$$
F_{R_j}(t) = 1 - \sum_{\ell=0}^{j-k-1} \binom{k-1+\ell}{\ell} t^{-d(k-1)}(1-t^{-d})^\ell \leq 1 - \sum_{\ell=0}^{j-k-1} \binom{k-1+\ell}{\ell} t^{-dk}(1-t^{-d})^\ell . \tag{62}
$$

Notice that $\binom{k-1+\ell}{\ell} t^{-dk}(1-t^{-d})^\ell$ is the pmf of negative binomial distribution $\text{NB}(k, 1 - t^{-d})$. Therefore, $F_{R_j}(t) \leq \mathbb{P}\{X \geq j - k\}$, where $X \sim \text{NB}(k, 1 - t^{-d})$. The mean and variance of $X$ are given by $\mathbb{E}[X] = (1 - t^{-d})k/(1 - (1 - t^{-d})) = (t^d - 1)k$ and $\text{Var}(X) = (1 - t^{-d})k/(1 - (1 - t^{-d}))^2 = (t^{2d} - t^d)k$. Therefore, by Chebyshev inequality, the tail probability is upper bounded by:

$$
\mathbb{P}\{X \geq j - k\} \leq \frac{\text{Var}(X)}{(j - k - \mathbb{E}[X])^2} = \frac{(t^{2d} - t^d)k}{(j - k - (t^d - 1)k)^2} = \frac{(t^{2d} - t^d)k}{(j - t^d k)^2} \leq 4t^{2d}k/j^2 , \tag{63}
$$

here we use the fact that $t \leq (j/2k)^{1/d}$ so $j - t^d k > j/2$. Therefore, $F_{R_j}(t) \leq 4t^{2d}k/j^2$ for $t > (j/2k)^{1/d}$. Combine the two cases and plug into (61), we obtain:

$$
\begin{aligned}
\mathbb{E}\Big| \frac{(\sum_{\ell=1}^{j} E_\ell)^{\alpha/d}}{(\sum_{\ell=1}^{k} E_\ell)^{\alpha/d}} \exp\{-\frac{(\sum_{\ell=1}^{j} E_\ell)^{2/d}}{2(\sum_{\ell=1}^{k} E_\ell)^{2/d}}\} \Big| &= \int_{t=1}^{\infty} 2t(t^2-1)e^{-t^2} F_{R_j}(t)dt \\
&\leq \int_{t=1}^{(j/2k)^{1/d}} 2t(t^2-1)e^{-t^2} \frac{4t^{2d}k}{j^2} dt + \int_{(j/2k)^{1/d}}^{\infty} 2t(t^2-1)e^{-t^2} dt \\
&\leq \frac{8k}{j^2} \int_{t=1}^{\infty} t^{2d+3} e^{-t^2} dt + 2\int_{(j/2k)^{1/d}}^{\infty} t^3 e^{-t^2} dt \\
&\leq \frac{8kC_d}{j^2} + 2\left( -\frac{1}{2} e^{-t^2}(t^2+1)\Big|_{(j/2k)^{1/d}}^{\infty} \right) \\
&= \frac{8kC_d}{j^2} + e^{-(j/2k)^{2/d}}((\frac{j}{2k})^{2/d}+1) \,,
\end{aligned}
\tag{64}
$$

where $C_d = \int_{t=1}^{\infty} t^{2d+3} e^{-t^2} dt$ is a constant only depend on $d$. Therefore, we can see that

$$
\mathbb{E}\Big| \frac{(\sum_{\ell=1}^{j} E_\ell)^{\alpha/d}}{(\sum_{\ell=1}^{k} E_\ell)^{\alpha/d}} \exp\{-\frac{(\sum_{\ell=1}^{j} E_\ell)^{2/d}}{2(\sum_{\ell=1}^{k} E_\ell)^{2/d}}\} \Big| = O(1/j^2).
\tag{65}
$$

So

$$
\mathbb{E}\|\tilde{T}_{\alpha,i}^{(m)}\| \leq \sum_{j=m+1}^{\infty} \mathbb{E}\Big| \frac{(\sum_{\ell=1}^{j} E_\ell)^{\alpha/d}}{(\sum_{\ell=1}^{k} E_\ell)^{\alpha/d}} \exp\{-\frac{(\sum_{\ell=1}^{j} E_\ell)^{2/d}}{2(\sum_{\ell=1}^{k} E_\ell)^{2/d}}\} \Big| \rightarrow 0 \,.
\tag{66}
$$

given $m \rightarrow \infty$ as $n \rightarrow \infty$.

## G  Proof of Theorem 2

The proposed estimator is a solution to a maximization problem $\widehat{a} = \arg\max_a \mathcal{L}_{X_i}(f_{a,X_i})$. From [21] we know that the maximizer is a fixed point of a series of non-linear equations of the form

$$
\begin{aligned}
\sum_{j\neq i} \frac{(X_j - X_i)^{\otimes\alpha}}{\rho_{k,i}^\alpha} K\Big(\frac{X_j - X_i}{\rho_{k,i}}\Big) \\
= n\rho_{k,i}^d e^{a_0} \int \frac{(u-X_i)^{\otimes\alpha}}{\rho_{k,i}^\alpha} K\Big(\frac{u-X_i}{\rho_{k,i}}\Big) e^{\langle u-x,a_1\rangle + \cdots + a_p[(u-x),\cdots,(u-x)]} \frac{1}{\rho_{k,i}^d} du
\end{aligned}
$$

for all $\alpha \in [p]$ where the superscript $\otimes\alpha$ indicates the $\alpha$-th order tensor product. From the proof of Theorem 1, specifically (48) and (50), we know that the left-hand side converges to a value that only depends on $k, d$ and $K$. Let's denote it by $S_\alpha(k) \in \mathbb{R}^{d^\alpha}$. We make a change of variables $\widetilde{a}_0 = a_0 + d\log\rho_{k,i} + \log n$ and $\widetilde{a}_\alpha = a_\alpha/\rho_{k,i}^\alpha$ for $\alpha \neq 0$. Then, in the limit of growing $n$, the above equations can be rewritten as

$$
S_\alpha(k,d,K) = e^{\widetilde{a}_0} F_\alpha(d,K,\widetilde{a}_1,\ldots,\widetilde{a}_p) \,,
\tag{67}
$$

for some function $F_\alpha$. Notice that the dependence on the underlying distribution vanishes in the limit, and the fixed point $\widetilde{a}$ only depends on $k, p, d$, and $K$. The desired claim follows from the fact that the estimate is $\lim_{n\rightarrow\infty} \widehat{f}_n(X_i) = \lim_{n\rightarrow\infty} e^{\widehat{a}_0} = \lim_{n\rightarrow\infty} A_{k,d,p,K}/(n\rho_{k,i}^d) = f(X_i)A_{k,d,p,K}C_d \lim_n 1/(C_d n\rho_{k,i}^d f(X_i)) = f(X_i)A_{k,d,p,K}C_d/\sum_{\ell=1}^{k} E_\ell$, and plugging in the entropy estimator $\widehat{H}(X) \rightarrow E_{X_i}[-\log f(X_i)] + B_{k,d,p,K}$.

In the case of the KL estimator, it happens that $S_0 = k$ and $F_0(d) = C_d$ such that $e^{\widetilde{a}_0} = k/C_d$, $e^{\widehat{a}_0} = f(X_i)k/(C_d\rho_{k,i}^d f(X_i)n)$ and $B_{k,d,p,K} = -\log k + E[\log(\sum_{\ell=1}^{k} E_\ell)] = -\log k + \phi(k)$.