[Reviews · NeurIPS 2016]

Reviewer 1

Summary

This paper unifies several previous approaches to the problem of density and entropy estimation from i.i.d. samples of a random variable. The authors derive new estimators and provide theoretical and experimental improvements over previous state-of-the-art (generalizing previous approaches to the problem). They accomplish this by reducing the data-dependent bandwidth of a local likelihood density estimator, and then proving that the increased bias is distribution independent and can therefore be subtracted off. Under some assumptions, the authors further can write this estimation bias in closed form.

Qualitative Assessment

The technical quality seems very good, worthy of an oral presentation. The authors should discuss the theoretical impact of truncation to $\log n$ nearest neighbors discussed on line 153-155. This is done to improve computational efficiency without discussion of how it affects the estimation. My only other technical comment is on the scaling parameters of order statistics. Lemma 3.1 assumes $m_n$ is order $\log n$, but in the experiments m=50,000 and n is either 1,000,000 (Table 1) or between 100 and 1500 (all other experiments). The authors should comment on the latter case where m >> n, and the convergence of their total variation bound as a function of m and n. The work seems significantly novel and can have quite a large impact since it generalizes and unifies several previous heuristic results. However, empirical performance of mutual information and entropy estimators are only shown on synthetic data. An unsupervised learning problem on real data would have shown the true impact of this approach to the machine learning community. Overall the paper is written well and explains the previous history and setup of the studied problems. There are a few grammatical errors and typos that do not detract from the overall presentation ("Despite the advantages of the LLDE, they require...", "k-NN method suffer from…”). Compound words like "log-likelihood" and "bottleneck" are split into two words when they are typically combined. Also, equation (2) cuts into the equation label. Some log-log plots only have logarithmic axes on the y axis. I enjoyed placing experimental results and figures in the relevant sections throughout. My only other suggestions would be to provide more intuition on the choice of exponential random variables/random vectors on the sphere, and to move some of the discussion in Section 6 to earlier in the paper.

Confidence in this Review

2-Confident (read it all; understood it all reasonably well)


Reviewer 2

Summary

This paper proposes a novel entropy and mutual information estimator that in a way combines kNN and kernels, by making the (local) kernel bandwidths dependent on kNN distances. This adaptive bandwidth selection results in s bias that the authors characterize analytically. Also, as opposed to a few other recent methods, this work establishes closed-form solution for local likelihood density estimation for certain choices of the local parametric form.

Qualitative Assessment

Recent work on entropy (and mutual information) estimation has shown that traditional plug-in estimators that rely on local uniformity assumption underperform under certain realistic conditions. Thus, recent efforts have proposed improved estimators that do not rely on local uniformity. This paper makes several novel and important contributions in that direction. First, it considers a local likelihood density estimation (LLDE) with exponential polynomial family, and establishes a closed form solution for polynomial degrees p=1,2. Remarkably, the results hold in arbitrary dimensions, which greatly improves on existing results (e.g., the practicality of LLDE in high dimensions was previously limited by the need to calculate high dimensional integrals). Second, the paper suggests a plug-in entropy estimator that uses LLDE with data-dependent bandwidth. As their main theoretical contribution, the authors derive a closed form expression of the bias term (resulting from the choice of local/adaptive bandwidth), and show that this term does not depend on data distribution. The authors also show that the above result holds for more general class of entropy estimators. And finally, the authors replicate the setup of KSG estimator in their framework by using correlated bandwidth, and show that it results in improved estimator for mutual information. This is an excellent paper overall, which advances state of the art in LLDE and entropy estimation. My main issue with the paper is the limited evaluation of the method against other approaches such as the ones proposed in [1,2,11], and which are known to significantly outperform the baselines considered by the authors. Also, all the experiments are for low-dimensional examples. Other comments: - Line 251: “Mutual information estimators have been recently proposed in [1, 2, 12], which claim to solve similar local likelihood maximizations as ours.” This language unnecessarily downplays the results from prior studies. While the solution provided here is better in certain ways, that does not invalidate the claims made in those studies. - Eq. 23: Should be Z-s instead of W-s

Confidence in this Review

2-Confident (read it all; understood it all reasonably well)


Reviewer 3

Summary

This paper studies the bandwidth selection problem for shannon entrap estimation using local like likelihood based density estimator. Based on an analysis of the asymptotic bias, the authors propose to use KNN bandwidth selector but instead of the more classical data-independent estimator.

Qualitative Assessment

My main concern for this paper is that the theoretical analysis does not provide any explicit rate of convergence (either in terms of MSE or in terms of high probability rate). This is a little surprising, since almost all the rest of the estimators have strong guarantees on the convergence rate.

Confidence in this Review

2-Confident (read it all; understood it all reasonably well)


Reviewer 4

Summary

The authors study the estimation of entropy and mutual information from samples. Their approach combines geometric and kernel-based ones: it uses a bandwidth choice of fixed k-nearest neighbor distances. Their new estimator has a bias that is universal in the sense that it is asymptotically independent of the underlying distribution; after subtracting this, the estimator becomes (asymptotically) unbiased. The authors provide extensive theoretical and practical results that support the validity and strength of the results,

Qualitative Assessment

The paper is generally well written and contains interesting results, both theoretically and experimentally. The idea of combining the two types of methods is nice and implemented well. The topic is certainly of interest to the NIPS community, and thus I believe that this contribution could be worthwhile to consider.

Confidence in this Review

2-Confident (read it all; understood it all reasonably well)


Reviewer 5

Summary

The paper studies nonparametric estimation of of differential entropy and mutual information and proposes a family of estimators based on resubstitution of a local likelihood density estimate, with a (fixed k) k-nearest neighbor bandwidth. This family includes the classic Kozachenko-Leonenko entropy estimator. Closed-form expressions for the estimators are derived for several cases of interest (Proposition 2.1). Results on the asymptotic distributions of k-NN distances (Lemma 3.1) are used to show that, after adding a bias correction (that can be estimated via Monte Carlo methods), the family of estimators is asymptotically unbiased (Theorem 2). In the particular case that variables have strongly functional relationships, it is suggested theoretically and empirically that the proposed estimators are a substantial improvement over the state-of-the-art.

Qualitative Assessment

--------------------------------------------------------------------------------- EDITS BASED ON AUTHOR RESPONSE AND DISCUSSION I still feel the paper has too many issues to be accepted. If the paper is accepted, I would VERY strongly urge the authors to consider some of the below changes: 1) Include empirical results on more diverse data. Except in Figure 3(b) (where results are inconclusive because the true value is not shown, and estimators converge to different values), experiments are all on Gaussian data. This makes them less convincing for two reasons: a) With p = 2 and a Gaussian kernel, the LLDE model class includes the true distribution (for finite n), and the local log-likelihood (which is the estimand for entropy) estimate suffers no smoothing bias. This is an unusual advantage for a nonparametric model, and eliminates the main source of bias in nonparametric density estimation. b) Detecting near-functional relationships (i.e., very strong dependencies) via mutual-information is mostly of interest for non-linear dependencies, which are not studied here. Hence, I'm worried the experiments were performed in a much easier setting than where they would be useful - I find the experiments in [2] much more convincing." 2) Explain how the current work fits into prior work and conceptual understanding (ignoring Section 6, for now): a) Discuss WHY the novel part of Lemma 3.1 (jointly uniform distribution of kNN directions) is relevant here (i.e., because the local Gaussian fit relies on kNN vectors rather than just distances), constrasting with, say Leonenko, Nikolai, Luc Pronzato, and Vippal Savani. "A class of Rényi information estimators for multidimensional densities." The Annals of Statistics 36.5 (2008): 2153-2182). b) Mention that Proposition 2.1 is based on a classic result. Otherwise, I think a reader might be quite surprised to read that a recent paper has found a new closed form for the classic Gaussian LLDE. c) Claims about superior performance over other estimators NEED TO BE SUPPORTED BY EXPERIMENTALLY! Either tone these down or justify them. d) The claims about boundary bias should be corrected, as discussed earlier: "Lines 125-145: I find it misleading to attribute the bias of the KL (and p = 1) estimator to the "boundary". Consider, for example, the distribution Beta(0.5, 0.5) (or any similar distribution on the interval [0, 1] whose maxima are on the boundary). A k-NN density estimator would experience the same source of bias on the INTERIOR of [0, 1]. The issue is actually that k-NN density estimators are biased at points of low density. (See Theorem 2 of Mack, Y. P., and Murray Rosenblatt. "Multivariate k-nearest neighbor density estimates." Journal of Multivariate Analysis 9.1 (1979): 1-15, in particular, the term ||H_f(x)||/(f(x))^(2/d) (similar terms appear in this paper's Lemma 3.1)." 3) Significantly shorten and simplify Section 6 (and the whole "breaking the bandwidth barrier theme"). As I understand Section 6 argues that kNN bandwidth is a good choice, with the reasoning: "For KDE resubstitution estimators, the bias is O(h^{2\beta} + 1/(nh^d)). With a nearest-neighbor bandwidth (O(roughly, h = O(n^{-1/d}), one can correct the bias to be O(h^{2\beta} + E_n) = O(n^{-2\beta/d} + E_n), where E_n is error due to the the bias correction being asymptotic, creating a consistent estimator with a smaller h^{2\beta} term." Note that we generally know nothing about E_n beyond E_n -> 0, so this provides NO THEORETICAL JUSTIFICATION FOR THIS BEING A GOOD IDEA. The sole exception is the KL estimator, because the bias correction is exact in the sense of point 8)a)) below (E_n = 0 for all n). Hence, Section 6 does not, to me, suggest that kNN distances are 'the "correct" mathematical choice for the purpose of estimating an integral functional such as the entropy.' 4) Fix Figure 4: Comparing to Silverman's rule of thumb in Figure 4 does not make sense, because there are much better established deterministic bandwidth choices for this problem. For a kernel density estimator (KDE), [9] and subsequent work suggest using a bandwidth of order O(n^{-1/(\beta + d)}), where \beta is the Holder exponent of the density. In this case, \beta = 0, suggests a bandwidth of O(n^{-1/2}), much smaller than O(n^{-1/5}). Thus, Figure 4 says more about the poor choice of deterministic bandwidth than about kNN bandwidth. 5) Include experiments in higher dimensions: Experiments are all in 2D. In high dimensions, nonparametric methods suffer because of their lack of structural assumptions (i.e., too large a model class). Since LLDE takes this even further than, say KDE, I'm skeptical that these estimators will work except in very low dimensions. 6) Contrary to line 157, I find the assumptions that ||H_f(x)||/(f(x))^3 and \|\nabla f(x)\|/f(x) are bounded quite strong, as they exclude the tail behavior of most standard parametric examples. 7) Explain the proof of Theorem 1: Specifically, on line 387 of the supplement, it is assumed that \bar h is bounded by 10^10 (this is later used to apply the dominated convergence theorem). I don't see why this holds. 8) Minor comments/questions: a)The paper does not quite generalize the KL estimator as claimed, as the latter has a slightly stronger property: the usual bias correction, in the sense described above, for the KL estimator is NOT asymptotic (it depends on n: \psi(n) - \log(n) + \log(k) - \psi(k); the value on line 229 is an asymptotic simplification). b) In Figure 3 (right), what is the true mutual information? The various estimators do not appear to converge to the same value. Also, what are the variances? Error bars would be nice. c) If the ideas of Section 6 can be shortened, I think it would be nice to discuss which aspects of the current work might generalize to other functionals. d) Discuss the regime of interest for these estimator (i.e., when the correlation is EXTREMELY strong (r > 0.999)). e) Discuss computational complexity of the proposed estimators. Note that k-d trees only work well in moderately low dimension, so this is a constraint on these (and most nonparametric) estimators in high dimension. f) At least suggest whether the variance vanishes asymptotically (e.g., whether the estimator is weakly consistent in the sense of converging in probability to the true value), even if it is not proven. While I agree this seems likely using, say the methods in Biau & Devroye (2015), I think it is not obvious, since the LLDE is not itself consistent. g) I would find it more informative if the error plots showed bias and variance separately (e.g., showing variance with error bars), rather than showing MSE, so that one could see which primarily contributes to the error, and how this varies with, e.g., sample size, dimension, etc.

Confidence in this Review

3-Expert (read the paper in detail, know the area, quite certain of my opinion)


Reviewer 6

Summary

The authors propose a new class of estimators for Shannon entropy (and Mutual Information (MI)). The estimators are density plug-in estimators based on the empirical average of the log of density estimates at points drawn from the underlying distribution. For density estimation, the authors use local likelihood density estimators (LLDE’s), which are higher-order generalizations of KDE and kNN density estimators.

Qualitative Assessment

The most potentially novel contribution of the paper is the discussion in section 6 where the authors show how adaptive bandwidths (i.e. k-nn) can be superior to fixed bandwidth (i.e. bandwidth is the same at all points). However, the discussion focuses on Shannon entropy only. If the results in this discussion can be applied to other distributional functionals (e.g. Renyi entropy and others), then it is indeed very novel. However, if it only applies to Shannon entropy, then it is quite limited. This similarly applies to all of the contributions. The authors should comment on the feasibility of extending these results to other distributional functionals. This is critical as all current state of the art estimators focus on more than just Shannon entropy. Another contribution of the paper is the result that the empirical average converges to the true entropy plus a constant term B(k,p,d, K) that is independent of the underlying distribution for any choice of kernel K, order p, and parameter k. If the term B(k,p,d) were to be distribution-dependent, then typically k is required to go to infinity (but at a slower rate compared to n) in order to ensure convergence, which results in slower rates of convergence. But because the authors show that B(k,p,d) is independent of the underlying density, they can estimate this quantity using Monte-Carlo methods, and subtract this constant term from their empirical average to derive an estimator that converges to the true entropy even with k fixed. Results of this form have been established before for specific choices of kernel K and order p. For example, in [6], the result is established for p = 0 and the choice of a step kernel. This paper shows that [6] (and other estimators) are a special case of this very general result they have derived, which is a nice result. Of course, this raises the question as to what the advantages are with using a estimator with different choice of kernel and higher values of p are relative to the existing estimators which correspond to p = 0. The authors claim that the advantage is that the finite sample performance of higher-order estimators is better, and mainly because they do better at the edges of the support of the density, where the higher order density estimators have lower bias by virtue of adapting to the contours of the distribution (see figure 1). They present experimental results, (but no theoretical results), and claim that the higher order estimators have better finite sample MSE compared to [6]. These are some of the issues with the paper: 1- In Section 2, the authors claim that higher order LLDE’s automatically correct for boundary bias. Do they mean that boundary bias is completely eliminated by using LLDE’s? Or do they mean that LLDE’s do a better job of accounting for some of the boundary bias but still do not completely eliminate it? If it is the former, then this is a very strong claim and the analysis in Section 2 does not convince me this is true. For instance, for samples drawn from a distribution with compact support, I believe the density estimates for points on the boundary of the support would be underestimated irrespective of the order of the LLDE. If it is the later, the authors should make this more clear in Section 2. 1.1- On a related note, figure 1(b) helps readers visualize a standard k-NN density estimate and understand why it suffers from boundary bias. But there is no illustration of the LLDE density estimator in this figure. Addition of this illustration would greatly hep readers get a sense for why LLDE’s better account for the boundary bias. 2- The flip side to using LLDE’s of higher order is that they are more expensive to implement. The authors attempt to address this by using subsets of samples for evaluating equation (10). It would be interesting to compare the computation time for lower order kernels like the one in [6] against the higher-order kernels to see what the trade-off is between MSE performance and computation time. 3-The simulations have some major flaws. The authors claim that their estimator outperforms the state of the art estimators. This claim is far too strong. First, they only compare to one state of the art estimator [5] and completely neglect the other state of the art estimators (in terms of MSE convergence rate), most especially ensemble methods (see [5R,6R] below). Another state of the art estimator is included in [11R]. I am also not convinced that the authors implemented the Von mises estimator of [5] correctly as the MSE is nonmonotonic as the sample size increases in Figure 2. How did the authors choose the bandwidth and what kernel did they use? The authors should provide more details on these experiments. Also, in figure 2(b), the rate at which the MSE of the KL estimator is decreasing in sample size is much faster than the rate at which the p=2 LNN estimator’s MSE is decreasing. If the figure were extrapolated beyond n=1500 samples, it seems like the KL estimator would start to outperform the LNN estimator. Figure 3(b) seems to convey a similar story that the small sample performance of the LNN-KSG estimator is much better compared to the standard KSG, but again it is not clear that the rate of convergence of LNN-KSG is better than KSG. This is a critical issue that must be studied more carefully, especially given that the results presented are asymptotic convergence results and not finite sample analysis results. Finally, the authors only focus on the very low dimensional case. I suspect that in the higher dimensional setting, the state of the art estimators would outperform the proposed estimator which limits its applicability. 4- In addition to the KL and von-moses estimators for entropy, the KSG estimator for MI, and the estimators mentioned above, there are several other estimators in literature for these quantities that are important and have been neglected by the author. See the references below. Some if not all of these references should be included, especially those with known convergence rates. 5- Although the paper in its current form makes significant theoretical contributions, it would definitely be of interest to extend the results to obtain theoretical results that describe finite sample performance of these estimators (or equivalently, obtain the rate of convergence of the estimators). For instance, see [3R, 5R, 6R, 7R, 11R], where rates of convergence are obtained for various entropy and MI estimators. See also [12R] where the convergence rates of the KL estimator are derived. 6- In this paper, results are derived only for estimation of Shannon entropy and MI. As mentioned above, estimators for other forms of entropy and MI are also of interest. 6.1- For instance, see [7R, 8R, 9R] for work on Renyi entropy and divergence estimation. It would be useful to investigate if similar results to the ones derived in this paper hold for these Renyi measures. It has already been shown in [9R] that a bias correction term exists for Renyi entropy estimators (the factor C_k in section 3.1 in [9R]) - does a similar term exist for higher order LLDE’s? 6.2-In addition to Renyi estimators, there has also been work on estimation of general functionals of densities - see [3R, 5R, 10R, 11R]. To the best of my knowledge, there has been no work on establishing that bias correction factors exist for estimators of these general functionals, and as a result, density plug-in estimators require that k also grow to infinity. It would be very interesting to see if the authors could extend their analysis to estimators for general functionals other than log(x) (Shannon) and x^{a-1} (Renyi), and comment on whether bias correction factors exist for estimators of these general functionals. 6.3- This reviewer would be interested in hearing the authors take on whether they think their analysis is general enough to accommodate these more general functionals, and if not, if they could comment on what makes the log(.) function critical to their analysis. In summary, I would not expect the authors to address all of these issues in the rebuttal. At a minimum, I would expect the authors to 1) address the issues wrt the boundary bias; 2) Provide more details on the experiments, especially when implementing the Von Mises estimator; 3) Either soften their language with regards to the experimental results (i.e. recognize that their approach only outperforms one state of the art estimator in rather limited circumstances and at lower dimensions) OR compare their estimator to other state of the art estimators and at higher dimensions; 4) Include at least the references where convergence rates are known; 5) Comment on the feasibility of extending this work to more general functionals. If these issues are addressed, then I would recommend the paper be accepted for publication at NIPS. If any of the other issues are addressed, the paper would be even stronger. Additional references: [1R] Pál, Dávid, Barnabás Póczos, and Csaba Szepesvári. "Estimation of Rényi entropy and mutual information based on generalized nearest-neighbor graphs." Advances in Neural Information Processing Systems. 2010. [2R] Beirlant, Jan, et al. "Nonparametric entropy estimation: An overview."International Journal of Mathematical and Statistical Sciences 6.1 (1997): 17-39. [3R] Estimating divergence functionals and the likelihood ratio by convex risk minimization XL Nguyen, MJ Wainwright, MI Jordan - IEEE Transactions on Information Theory, 2010 [4R] Póczos, Barnabás, Liang Xiong, and Jeff Schneider. "Nonparametric divergence estimation with applications to machine learning on distributions."arXiv preprint arXiv:1202.3758 (2012). [5R] Moon, Kevin, and Alfred Hero. "Multivariate f-divergence estimation with confidence." Advances in Neural Information Processing Systems. 2014. [6R] Sricharan, Kumar, Dennis Wei, and Alfred O. Hero. "Ensemble estimators for multivariate entropy estimation." IEEE Transactions on Information Theory59.7 (2013): 4374-4388. [7R] Krishnamurthy, Akshay, et al. "Nonparametric Estimation of Renyi Divergence and Friends." ICML. 2014. [8R] Hero, Alfred O., et al. "Applications of entropic spanning graphs." IEEE signal processing magazine 19.5 (2002): 85-95. [9R] Leonenko, N., L. Prozanto, and V. Savani (2008), A class of r´enyi information estimators for multidimensional densities, Annals of Statistics, 36, 2153–2182. [10R] Sricharan, Kumar, Raviv Raich, and Alfred O. Hero. "Estimation of nonlinear functionals of densities with confidence." IEEE Transactions on Information Theory 58.7 (2012): 4135-4159. [11R] Singh and Poczos, "Exponential Concentration Inequality of a Density Functional Estimator," NIPS 2014. [12R] Biau and Devroye, "Entropy Estimation," In Lectures on the Nearest Neighbor Method," pp 75-91, Springer, 2015. EDIT: Based on the comments of other reviewers, I have downgraded my score for the novelty of this paper. I think that the novelty of theorems 1 and 2 is sufficient to maintain the score at a "3". However, I think that the authors should give more reference to prior work in the other less novel results as Reviewer 4 discussed. I think that this and toning down their claims on the experiments is critical (as well as referencing more prior work in general). Some of the other issues brought up by the other reviewers also lead me to leave the technical quality at a "2". These issues further seem to suggest that the authors' approach is mostly limited to Shannon entropy. If the authors had shown broader applicability to other functionals, I would have looked more favorably upon the paper despite these issues.

Confidence in this Review

2-Confident (read it all; understood it all reasonably well)